# Windows of developmental sensitivity to social media

Amy Orben [1✉], Andrew K. Przybylski[2], Sarah-Jayne Blakemore [3,4] & Rogier A. Kievit [5,1]

The relationship between social media use and life satisfaction changes across adolescent development. Our analyses of two UK datasets comprising 84,011 participants (10–80 years old) find that the cross-sectional relationship between self-reported estimates of social media use and life satisfaction ratings is most negative in younger adolescents. Furthermore, sex differences in this relationship are only present during this time. Longitudinal analyses of 17,409 participants (10–21 years old) suggest distinct developmental windows of sensitivity to social media in adolescence, when higher estimated social media use predicts a decrease in life satisfaction ratings one year later (and vice-versa: lower estimated social media use predicts an increase in life satisfaction ratings). These windows occur at different ages for males (14–15 and 19 years old) and females (11–13 and 19 years old). Decreases in life satisfaction ratings also predicted subsequent increases in estimated social media use, however, these were not associated with age or sex.

[1] MRC Cognition and Brain Sciences Unit, University of Cambridge, Cambridge, UK. [2] Oxford Internet Institute, University of Oxford, Oxford, UK.
[3] Department of Psychology, University of Cambridge, Cambridge, UK. [4] Institute of Cognitive Neuroscience, University College London, London, UK.
[5] Cognitive Neuroscience Department, Donders Institute for Brain, Cognition and Behavior, Radboud University Medical Center, Nijmegen, The Netherlands.
✉email: amy.orben@mrc-cbu.cam.ac.uk

Technological innovations have shaped the ways in which we connect with each other[1]. Yet the recent adoption of social media has fundamentally transformed how humans spend their time, portray themselves and communicate. The repercussions of such changes have induced widespread concern[2–5]. Yet there is still considerable uncertainty about how social media use relates to well-being. Meta-analyses have identified small or negligible negative links between social media use and well-being[6,7], while experimental evidence is mixed[8,9]. Longitudinal observational studies that have investigated the predictive relationships between social media use and well-being have found that they are either reciprocal[10], only present in a certain direction or sex[11] or not present at all[12]. The lack of concrete evidence is an issue routinely highlighted by academics[2,3], medical professionals[13,14], and policymakers[15,16].

Much needed progress in understanding how social media use affects well-being could be made by studying the phenomenon through a developmental lens, acknowledging that developmental processes can alter our sensitivity to both the positive and negative impacts of social media[17]. One such developmental stage is adolescence, which spans 10–24 years[18], and represents a period of profound biological, psychological, and social development. It has been proposed that substantial biological changes in the social brain make adolescence a sensitive period for social development[19], self-perception, and social interaction[20]. Adolescence is also a time of cognitive development, especially in domains such as emotional regulation, planning, and response inhibition[21]. In parallel, most adolescents go through major sociocultural changes and life events such as moves from school to university or work. Such biological, psychological, and social changes magnify the influence of an adolescent's social environment and make them more attuned to how they are perceived by peers and the broader community. It is therefore plausible that these processes heighten adolescents' sensitivity to the interactive, communicative, and self-portraying nature of social media[22], a technology they use more extensively than other age groups[23,24].

It is possible that sensitivity to social media does not remain uniformly elevated throughout adolescence, given the diverse and protracted developmental processes experienced during this time[19]. Periods of increased sensitivity to social media are likely to occur specifically in parallel to relevant developmental changes. The strength of the statistical relationship between social media use and outcomes such as well-being might therefore not only be expected to vary between adolescence and other life stages, but also across adolescent development. To locate such developmental windows of sensitivity to social media, it is necessary to explicitly account for the developmental stage in research design and analysis strategy. By extending past work to account for age, as a proxy for development, and to encompass the whole adolescent range, this study tests for hypothesized developmental windows of social media sensitivity when stronger links between social media and well-being emerge at specific ages.

In examining adolescence, it is additionally important to consider how sex interacts with the developmental stage. While some adolescent developmental processes (whether they be biological, cognitive, or social) are similar in both character and timing across sex, others show variation that needs to be accounted for. For example, females experience pubertal bodily changes earlier than do males, which can provoke further downstream social changes[20]. Further, female life satisfaction drops earlier in adolescence[25,26] and the risk of certain mental health problems such as depression, self-harm, and eating disorders is higher in adolescent females than in males[27]. Research has highlighted differences between males and females in the links between social media use and well-being in adolescence in a small subset of the analyzed data[28], or other datasets[10,29–33]. This study therefore also examines potential sex differences in how social media use relates to well-being across adolescent development.

To investigate the existence of developmental windows of sensitivity to social media, we first use cross-sectional data to examine whether adolescence might represent a period during which the association between well-being and social media is different in comparison to other life stages, and if differences between males and females are present during this time. Using longitudinal data, we then test the idea that there exist sex-specific windows of sensitivity to social media during adolescence itself.

## Results and discussion

To address the first research question, we analyzed the UK Understanding Society household panel survey that includes 72,287 10–80-year-old participants surveyed up to seven times each between 2011 and 2018[34], correlating a single-item life satisfaction measure and participant estimates of how much time they spend using social media on a typical day (raw data plot: Fig. 1; extended plot: Supplementary Fig. 1). We also tested the robustness of these relations using—both linear and quadratic— terms of estimated social media use to predict life satisfaction ratings while adding control variables of log household income, neighbourhood deprivation (measured using the Index of Multiple deprivation), and year of data collection (Supplementary Fig. 2).

While it is important to note that responses to, and conceptualizations of, life satisfaction might be qualitatively different across the lifespan, our results showed that the relationship between estimated social media use and life satisfaction ratings varied substantially by age. Although the relationship fluctuates to a certain extent across the lifespan, for example, it is more negative in males aged 26–29 years compared to males aged 22–25 years, the most substantial negative relations were found in adolescence (Supplementary Fig. 2). This finding, combined with our developmental interest in adolescence, the fact that social media use is heightened in this age group, and the nature of the data (annual longitudinal social media measures being available only for those aged 10–21), motivated us to focus on adolescence as the age group of interest throughout this study.

The relations in the raw data differed when comparing younger (10–15 years) and older adolescents (16–21 years; for more information about these age categories see methods). There was a pronounced inverted U-shaped curve in older adolescence, indicating that those who estimated they engaged in very low or very high social media use reported lower life satisfaction ratings than those who estimated that they used between 'less than an hour' and '1–3 h' of social media a day (i.e., response options '2' and '3'). This pattern in between-person associations, where those participants who use the least or the most social media also report lower well-being ratings, has been previously termed the 'Goldilocks hypothesis' (i.e., the concept that too much and too little digital technology use might be suboptimal)[35].

Younger adolescents demonstrated a different pattern of between-person associations than older adolescents: the relationship was more linear and showed more prominent differences between males and females (Fig. 1; top). Specifically, there was no evidence in this age range for the 'Goldilocks hypothesis,' as those who reported very little social media use did not routinely score lower on life satisfaction than their peers who reported slightly higher social media use. Further, females reporting very high social media use scored substantially lower on life satisfaction than males. This difference between males and females is statistically supported by an Akaike weights procedure[36], which allows us to quantify evidence ratios between two more models,

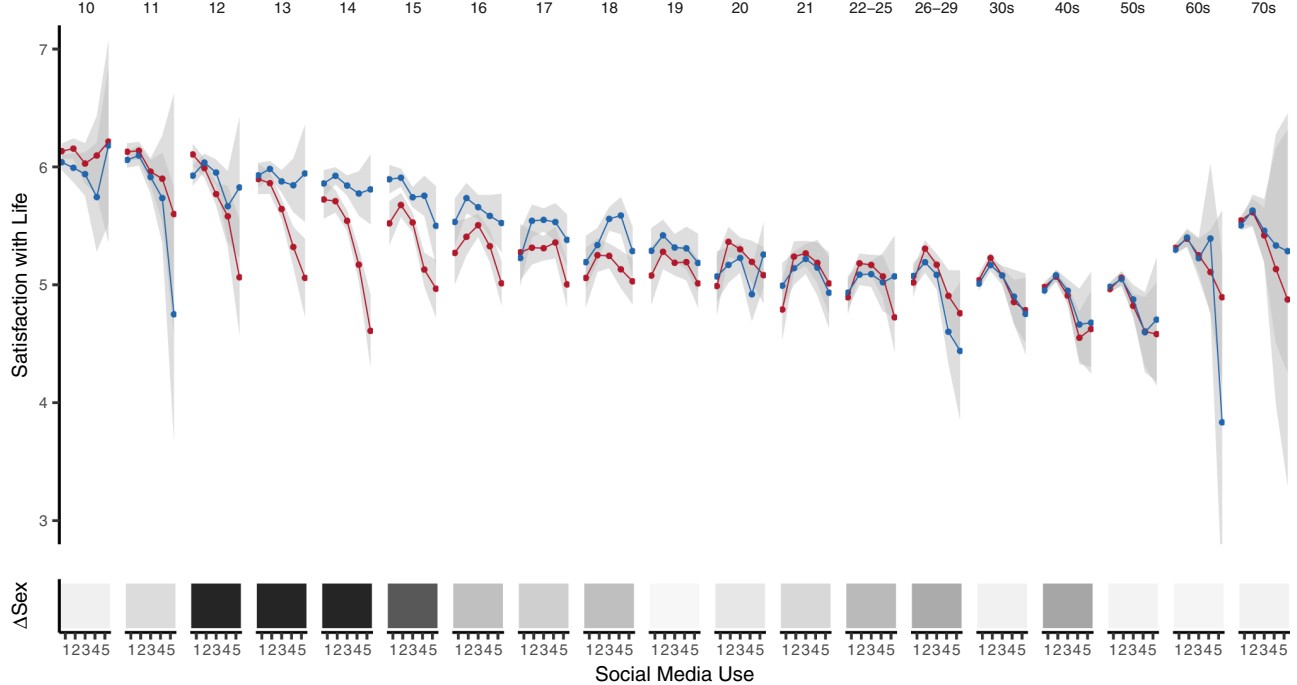

**Fig. 1 Estimated social media use and life satisfaction ratings across the lifespan.** Top: The cross-sectional relationship between estimated social media use and a one-item life satisfaction measure for 72,287 UK participants between the age of 10 and 80 years. The results are split by age and self-report sex: females = red, males = blue. The 95% confidence intervals represent the lower and upper Gaussian confidence limits around the mean based on the t-distribution. Bottom: Shading of each rectangle represents calculated AIC weights, i.e., whether a model relating estimated social media use and ratings of life satisfaction that takes into account a possible sex difference is more likely to represent the data than a model that does not take into account sex: darker shade = model with sex differences is more likely. It should be noted that as high levels of social media use are very rare in the youngest and oldest age groups present in the data (e.g., ages 10, 11, and 60+, Supplementary Fig. 1), one cannot evaluate functional form in these groups. Further, as most participants were measured multiple times, more than one data point per participant will appear in this graph. Source data for this figure are provided as a Source Data file.

provided the models are nested. Doing so, we show that models associating estimated social media use and life satisfaction ratings while differentiating for self-reported sex (and controlling for household income, neighbourhood deprivation, and year of data collection) are more likely to be the best models of the data between the ages of 12 and 15 (Akaike weights ratios ranging from 5:1 to over 6,000,000:1 in favour of the model differentiating males and females). In contrast, models differentiating for sex were not more likely to be the best models of the data at other ages (Fig. 1, bottom; Supplementary Fig. 3). These analyses demonstrated that the between-person association linking estimated social media use to ratings of life satisfaction was more negative in adolescents compared with most other age groups. Further, they showed that adolescence is unique due to the prominent sex differences on the cross-sectional links between estimated social media use and ratings of life satisfaction that are not evident at most other ages.

We supplemented these cross-sectional analyses by examining the longer life satisfaction questionnaire given to 10–15-year-olds in Understanding Society, and 13–14-year-olds in the Millennium Cohort Study (Fig. 2): this questionnaire asked about satisfaction with appearance, friends, family, school, schoolwork, and life. We found no evidence that a specific sub-component of life satisfaction was the lone driver of the sex differences found in Fig. 1. In Understanding Society, sex differences were found predominantly for satisfaction with life and appearance (Akaike weight of model with sex difference compared to the model without sex difference: satisfaction with life 71.9%, appearance 69.1%, family 53.8%, friends 59.1%, school 51.0%, schoolwork 35.0%; the percentage shows how much more likely a model

including a sex difference is the best model of the data compared to a model without a sex difference). In the Millennium Cohort Study, the models differentiating for sex were more likely to be the best model of the data for all measures (Akaike weight of model with sex difference compared to model without sex difference: satisfaction with life 100%, appearance 100%, family 100%, friends 99.7%, school 100%, schoolwork 91.2%). Further analyses using a broader range of mental health questionnaires available for these sample participants can be found in Supplementary Methods 1, Supplementary Results 1, and Supplementary Figs. 4–6. The limited 10–15-year age range available for these measures did not allow us to compare these adolescents with other age groups.

To address our second research question, and locate developmental windows of sensitivity to social media that are hypothesized to emerge across adolescence, we used methods that test within-person changes and differences over time[37,38]. While the previous analyses allowed us to examine whether adolescence potentially represents a period of heightened sensitivity to social media in comparison to other age groups, cross-sectional differences cannot be used to map detailed developmental processes. Longitudinal models such as the Random-Intercept Cross Lagged Panel Model (RI-CLPM) need to be used to take into account these dynamic and reciprocating relations between social media use and ratings of life satisfaction[10]. We estimated these models with robust maximum likelihood to account for deviations from the assumption of multivariate normality and ensure all model comparisons are performed on nested models, and use Full information Maximum Likelihood to account for missingness which may vary systematically across measured covariates and

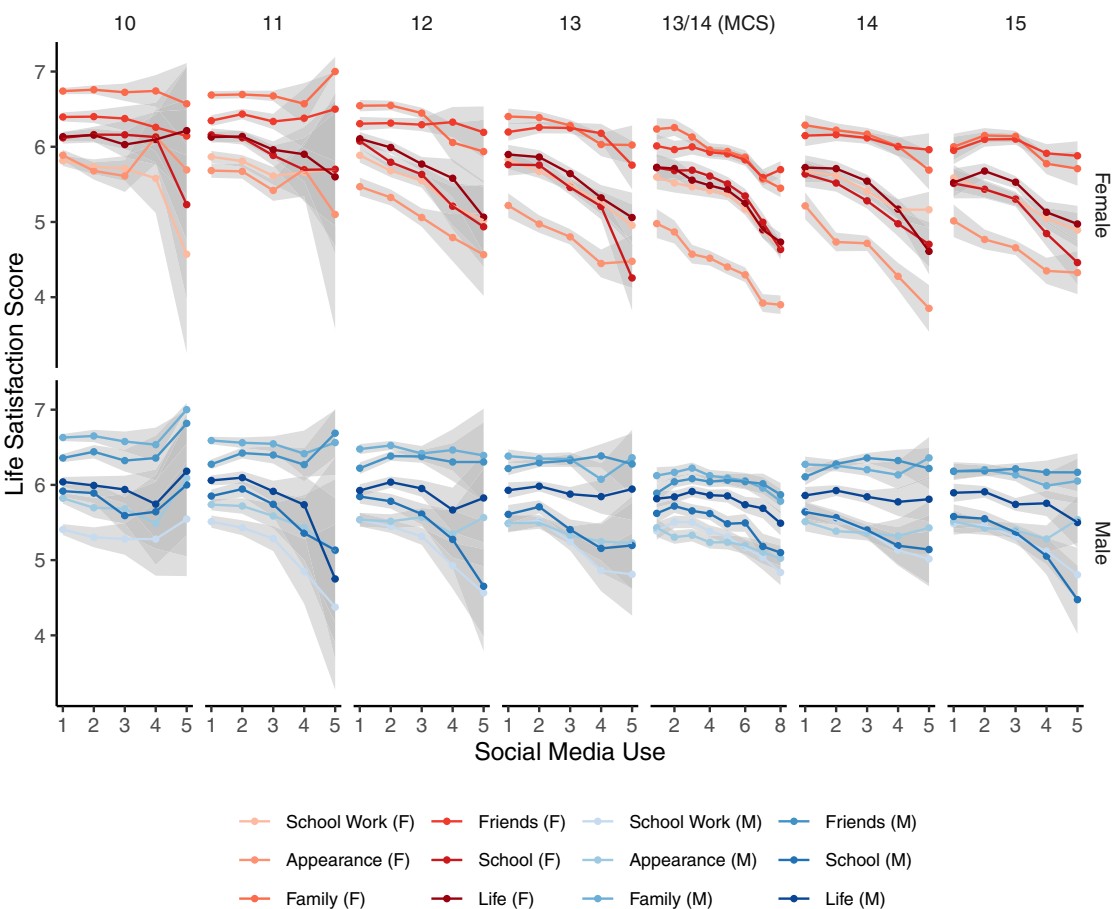

**Fig. 2 The cross-section relationship between social media use and six different life satisfaction measurements (ages 10–15).** The Figure shows the cross-sectional relation between estimated social media use and six life satisfaction measures at ages 10–15 (Understanding Society dataset, US; 10,019 participants and 24,698 measurement occasions) and ages 13–14 (Millennium Cohort Study dataset, MCS; 11,724 participants). Specifically, it displays the cross-sectional correlation between estimated social media use and raw scores of sub-components of life satisfaction (satisfaction with school work, appearance, family, friends, school, and life). The relationships are presented separately for males and females. The 95% confidence intervals represent the lower and upper Gaussian confidence limits around the mean based on the t-distribution. Source data for this figure are provided as a Source Data file.

model variables. We applied this modelling framework to data provided by 10–21-year-olds in Understanding Society, who completed estimated social media use and life satisfaction measures annually for up to seven waves (17,409 participants; for additional longitudinal analyses of component life satisfaction measures only available for 10–15-year-olds see Supplementary Fig. 7)[39,40]. As the outcomes of these longitudinal models depend largely on the time-interval between observations[41,42], the annual nature of the data needs to be considered. It is likely that studies on different time frames would show different results and/or reflect distinct mechanisms and processes.

Using longitudinal modelling, we can address the question of whether the present use of social media has consequences for future life satisfaction—and vice versa. Do people feel better, or worse, after periods of heightened social media use? Conversely, do people use more, or less, social media after periods of higher life satisfaction? Specifically, the RI-CLPM model allowed us to focus on whether an individual's deviation from their expected level of a certain variable $y$ (e.g., ratings of life satisfaction) can be predicted from their prior deviation from their expected scores in another variable $x$ (e.g., estimated social media use), while controlling for the structural change in $y$ (e.g., ratings of life satisfaction); and vice versa. We fit an initial model that allowed both cross-lagged paths to vary across age and sex ($\chi^2$ (434) = 1216.29, $p < 0.001$, RMSEA = 0.014, [0.013, 0.015], CFI = 0.944, SRMR =

0.072; Full Informational Maximum Likelihood estimation; two-tailed test). The control variables included in this model are time-invariant mean log household income and Index for Multiple Deprivation with freely estimated effects at different ages, to account for the socioeconomic status of both the family and their immediate environment. All RI-CLPM significance tests are two-sided. While the cross-lagged paths were predominantly non-significant, there are specific developmental windows where the data suggests estimated social media use and ratings of life satisfaction do predict each other—and these ages differed for males and females (Fig. 3).

Before interpreting these potential windows of sensitivity, we examined whether they were statistically robust via statistical model comparison procedures. A model constraining the path of life satisfaction ratings predicting estimated social media use to be constant across age and sex, while allowing the path of estimated social media use predicting ratings of life satisfaction to vary, was more likely to be the best model for the data compared to other model constraints (Akaike weights procedure: 99.2%, see Supplementary Fig. 8; model fit for best fitting model: $\chi^2$ (455) = 1234.736, $p < 0.001$, RMSEA = 0.014, [0.013, 0.015], CFI = 0.945, SRMR = 0.072).

We, therefore, examined the paths of this model in more detail (both social media use predicting life satisfaction, split by age and sex, and life satisfaction predicting social media use, not split by

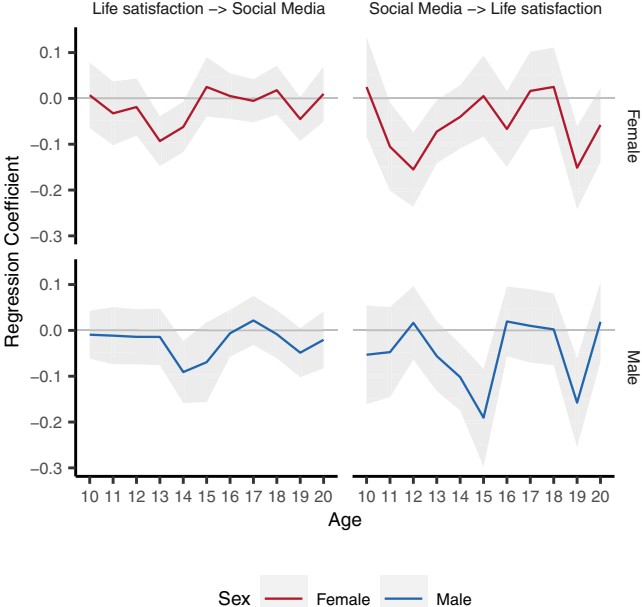

**Fig. 3 Results from Random Intercept Cross-Lagged Panel Model (RI-CLPM) of estimated social media use and life satisfaction for 17,409 participants of the Understanding Society dataset aged 10–21 (52,556 measurement occasions).** Results from both cross-lagged paths of a RI-CLPM where those paths were free to vary across age/sex. Results are unstandardized and split by path (left: deviations from expected ratings of life satisfaction at that age predicting deviations from expected social media use one year later; right: deviations from expected social media use at that age predicting deviations from expected ratings of life satisfaction one year later) and sex (female = top/red, male = bottom/blue). The ribbon represents the 95% Confidence Interval around the point estimate. All tests are two-sided. Source data for this figure are provided as a Source Data file.

age and sex, Table 1). The effect sizes of both paths were small in magnitude[43], something that has been discussed in previous work[3,10]. However, as dynamic effects can amplify over developmental time[44], they are nonetheless worth understanding and addressing with care. The constrained cross-lagged path of life satisfaction ratings predicting estimated social media use was negative, meaning that, for example, if an individual scored lower than their expected value of life satisfaction ratings in one year this predicted a positive change from their expected estimated social media use one year later (or vice versa, $b = -0.02$ [$-0.03$, $-0.01$], se = 0.007, $\beta = -0.02$–0.03, $p = 0.004$). The other cross-lagged path, linking social media use to life satisfaction, only showed a statistically significant negative link at specific ages, depending on sex (Fig. 4). This supports previous findings that the relationship between social media use and life satisfaction is bidirectional in nature[10], and provides evidence for the hypothesis that the impact of social media on individuals varies depending on how old they are, as well as their sex. We will focus on these differences in the next section, but wish to emphasise that this focus does not mean that the path linking life satisfaction ratings to estimated social media use one year later is unimportant.

Specifically, for females, we observed a window of sensitivity to social media between the ages of 11 and 13, when increases in estimated social media use from expected levels predicted a decrease in life satisfaction ratings from expected levels one year later (Fig. 4, top; age 11: $b = -0.11$ [$-0.21$, $-0.02$], se = 0.05, $\beta = -0.09$, $p = 0.020$, age 12: $b = -0.14$ [$-0.22$, $-0.07$], se = 0.04, $\beta = -0.12$, $p < 0.001$, age 13: $b = -0.08$ [$-0.15$, $-0.01$], se =

0.03, $\beta = -0.07$, $p = 0.019$). For males a similar window was in evidence at ages 14 and 15 (Fig. 4, bottom; age 14: $b = -0.10$ [$-0.17$, $-0.03$], se = 0.04, $\beta = -0.10$, $p = 0.005$, age 15: $b = -0.18$ [$-0.29$, $-0.08$], se = 0.05, $\beta = -0.12$, $p = 0.001$). Speculatively, the sex difference in timing suggested that early increases in sensitivity to social media might be due to maturational processes such as puberty, which occur earlier in females compared to males[20]. A later increase in sensitivity to social media, which was present at age 19 for both sexes, suggested a different underlying process may be present in late adolescence (Fig. 4; females: $b = -0.16$ [$-0.25$, $-0.07$], se = 0.05, $\beta = -0.13$, $p < 0.001$, males: $b = -0.16$ [$-0.26$, $-0.07$], se = 0.05, $\beta = -0.13$, $p = 0.001$). Particularly, the lack of sex differences indicated a process that similarly affects both males and females. Speculatively, this might be related to changes in the social environment such as a move away from home and subsequent disruptions in social networks. However, these explanations cannot be tested directly in this dataset and require further targeted investigation using data containing pubertal and social measurements.

The study has multiple limitations that need to be considered. First, to interpret the parameters from our analyses as estimates of causal effects one would need to adopt the following assumptions: (a) there are no time-varying unobserved confounders that impact the relation between social media use and life satisfaction; (b) the model adequately accounts for unobserved time-invariant confounding through the inclusion of a random intercept; (c) there is no measurement error in the variables; (d) the time interval between studies (one year) is the right length to capture the effects of interest; and (e) the bidirectional links estimated by our longitudinal model are linear in nature. Only if these assumptions are met can this observational study be said to capture the causal effects between social media and life satisfaction. Second, the data are self-report and therefore only allow inferences about the impact of self-estimated time on social media, rather than objectively measured social media use.

The findings reported here may enable investigation of potential mechanisms of interest, for example, in datasets with pubertal or additional social measurements. One could also carry out more targeted investigations, for example, by examining the mental health measures only completed by select age ranges in the datasets (e.g., ages 10–15, displayed in Supplementary Fig. 4) to understand how they interrelate over the longitudinal time frame. Furthermore, the cross-sectional relation between social media use and life satisfaction ratings showed differences across the whole life span, e.g., in early adulthood and old age. Future work may use a similar approach to investigate interactions in older age groups with a suitably rich sample.

This study provides evidence for age- and sex-specific windows of sensitivity to social media use in adolescence. Past research has demonstrated that each life stage exhibits its own unique trajectories, goals, and influences[19,45–47], and activities can therefore differ from being of no impact, to being adaptive or maladaptive depending on which developmental period one examines[46,48]. This applies to the activity of using social media, just as it applies to other activities such as exercise or drinking alcohol. While the results support past longitudinal work finding bidirectional influences between social media use and life satisfaction[10], it also goes beyond that to suggest that such influences vary, potentially due to concurrent developmental processes.

While the windows of sensitivity to social media are prominent in aggregate, they will most probably meaningfully differ across individuals, as each person's sensitivity is further influenced by a wide range of individual, peer, and environmental dynamics. Additionally, the types of social media use individuals are engaged in will add further variance to this complicated dynamic[49]. Our work, therefore, opens the door to new

**Table 1 Results of the best fitting Random-Intercept Cross Lagged Panel Model examining the bidirectional links between life satisfaction and social media use across ages 10–21 years.**

|  | b (Regression coefficient) | Standard error | β (Standardized effect size) | p |
|---|---|---|---|---|
| Life satisfaction –> social media use |  |  |  |  |
| Invariant across age and sex | −0.02 [−0.03, −0.01] | 0.007 | −0.02-0.03 | 0.004 |
| Social media use –> life satisfaction (female) |  |  |  |  |
| Age 10 | 0.02 [−0.09, 0.12] | 0.05 | 0.01 | 0.753 |
| Age 11 | −0.11 [−0.21, −0.02] | 0.05 | −0.09 | 0.020 |
| Age 12 | −0.14 [−0.22, −0.07] | 0.04 | −0.12 | 0.000 |
| Age 13 | −0.08 [−0.15, −0.01] | 0.03 | −0.07 | 0.019 |
| Age 14 | −0.04 [−0.11, 0.03] | 0.03 | −0.04 | 0.215 |
| Age 15 | 0.01 [−0.07, 0.10] | 0.04 | 0.01 | 0.784 |
| Age 16 | −0.07 [−0.15, 0.01] | 0.04 | −0.06 | 0.080 |
| Age 17 | 0.00 [−0.08, 0.08] | 0.04 | 0.00 | 0.937 |
| Age 18 | 0.02 [-0.06, 0.10] | 0.04 | 0.02 | 0.642 |
| Age 19 | −0.16 [−0.25, −0.07] | 0.05 | −0.13 | 0.000 |
| Age 20 | −0.06 [−0.14, 0.03] | 0.04 | −0.05 | 0.181 |
| Social media use –> life satisfaction (male) |  |  |  |  |
| Age 10 | −0.06 [−0.16, 0.05] | 0.05 | −0.04 | 0.291 |
| Age 11 | −0.05 [−0.15, 0.04] | 0.05 | −0.04 | 0.275 |
| Age 12 | 0.02[−0.06, 0.09] | 0.04 | 0.01 | 0.708 |
| Age 13 | −0.05[−0.12, 0.03] | 0.04 | −0.04 | 0.202 |
| Age 14 | −0.10 [−0.17, −0.03] | 0.04 | −0.10 | 0.005 |
| Age 15 | −0.18 [-0.29, −0.08] | 0.05 | −0.12 | 0.001 |
| Age 16 | 0.02 [−0.05, 0.10] | 0.04 | 0.02 | 0.551 |
| Age 17 | 0.01 [−0.07, 0.09] | 0.04 | 0.01 | 0.806 |
| Age 18 | −0.00 [-0.08, 0.08] | 0.04 | −0.00 | 0.956 |
| Age 19 | −0.16 [−0.26, −0.07] | 0.05 | −0.13 | 0.001 |
| Age 20 | 0.02 [−0.06, 0.11] | 0.04 | 0.02 | 0.614 |

The coefficients include 95% confidence intervals around the estimate.

theoretically informed approaches for studying how this increasingly pervasive technology impacts our population, by focusing on the within-person dynamics where effects actually unfold. In particular, an understanding of what neurodevelopmental, pubertal, cognitive, and social changes underlie developmental windows of sensitivity to social media, and how these are impacted by individual differences, could pave pathways for targeted interventions that address the negative consequences of social media while also promoting its positive uses. This will ultimately enable academic research to help inform critical policies, interventions, and conversations concerning adolescent well-being in the digital age.

## Methods

**Ethical approval**. The University of Essex Ethics Committee has approved all data collection on the Understanding Society main study and innovation panel waves, including asking consent for all data linkages except to health records. Ethical approval for the Millennium Cohort Study was given by the UK National Health Service (NHS) London, Northern, Yorkshire, and South-West Research Ethics Committees (MREC/01/6/19, MREC/03/2/022, 05/MRE02/46, 07/MRE03/32). No additional ethical approval was needed for this study.

**Datasets**. The study analyzed the Understanding Society dataset and the Millennium Cohort Study. The Understanding Society dataset is a longitudinal study following approximately 40,000 British households[34]. The study sample is designed to be representative of the UK population[50]. Started in 2009, its annual waves of data collection each span two years; we used 7 waves of data from between 2011 and 2018 released in February 2020 (the two first waves were excluded as parts of the sample were not asked to complete social media related questions). All household members between 10 and 15 years (whom we label here as 'younger adolescents') filled out a younger adolescent survey, while those 16 and over filled out an adult survey. 16–21-year-olds (whom we label here as 'older adolescents') further completed a short supplement with additional questions. Participants were incentivised with an unconditional £10 gift voucher at invitation to the study, furthermore adults had another £10 incentive if they completed the survey <5 weeks (if they received a web questionnaire). 10–15-year-olds were incentivised with a voucher. 16-year-olds were entered into a prize draw to win an iPad. Members of households that did not complete the previous wave were given an

incentive of £20. Incentivisation procedures changed slightly between waves and are detailed in the Understand Society fieldwork documentation. Oral consent was provided by parents and adolescents.

The Millennium Cohort study is a birth cohort study of a sample of around 11,000 young people born between September 2000 and January 2001[51]. The study over-sampled some parts of the population, for example, children from ethnic minorities (e.g., Indian, Pakistani, Bangladeshi, Caribbean, and African in the ethnic minority boost sample). In this study, we only used the wave of data collected in 2015, when the majority of respondents were 13 or 14 years old. This made the two datasets comparable (e.g., in terms of the prevalence and use of social media) to subsections of our Understanding Society sample. Participants were provided with a 'Participant Pack' (i.e., leaflet, membership card, key ring, travel card holder, and notebook) as incentivisation. Informed written consent was provided by parents and oral consent was provided by adolescents.

**Measures**

*Understanding society*. For the core analyses of the Understanding Society study, we examined life satisfaction ratings, estimated social media use, age, and sex measures derived from both younger adolescent, older adolescent, and adult surveys. We further supplemented these measures with two control variables—household income and Index of Multiple Deprivations—and a range of additional well-being and mental health questionnaires. To measure life satisfaction, younger adolescent survey respondents were asked to respond, "which best describes how you feel about your life as a whole?" (visual analogue scale ranging from 1 = very happy smiley face to 7 = very sad smiley face; written explanation: "1 is completely happy and 7 is not at all happy"; scale reversed so that higher scores indicate higher life satisfaction). They were asked the same question about how they feel about their school work, appearance, family, friends, and school. Adults and older adolescents were asked to "please select the answer which you feel best describes how dissatisfied or satisfied you are with the following aspects of your current situation… your life overall" (1 = completely dissatisfied to 7 = completely satisfied). Due to previous work showing that variables containing five or more categories can be treated as continuous with negligible drawbacks, we treated both measures as continuous[52].

The life satisfaction measures were different in wording and response options for the younger adolescent and older adolescent/adult survey to accommodate age differences, for example by including smileys for young adolescents. An irregularity in Understanding Society fieldwork provided us with an opportunity to test that they were not qualitatively different. Due to a lag in when questionnaires were issued into the field and completed by the participants, 37 16-year-olds mistakenly took the younger adolescent survey while 10 15-year-olds mistakenly took the older

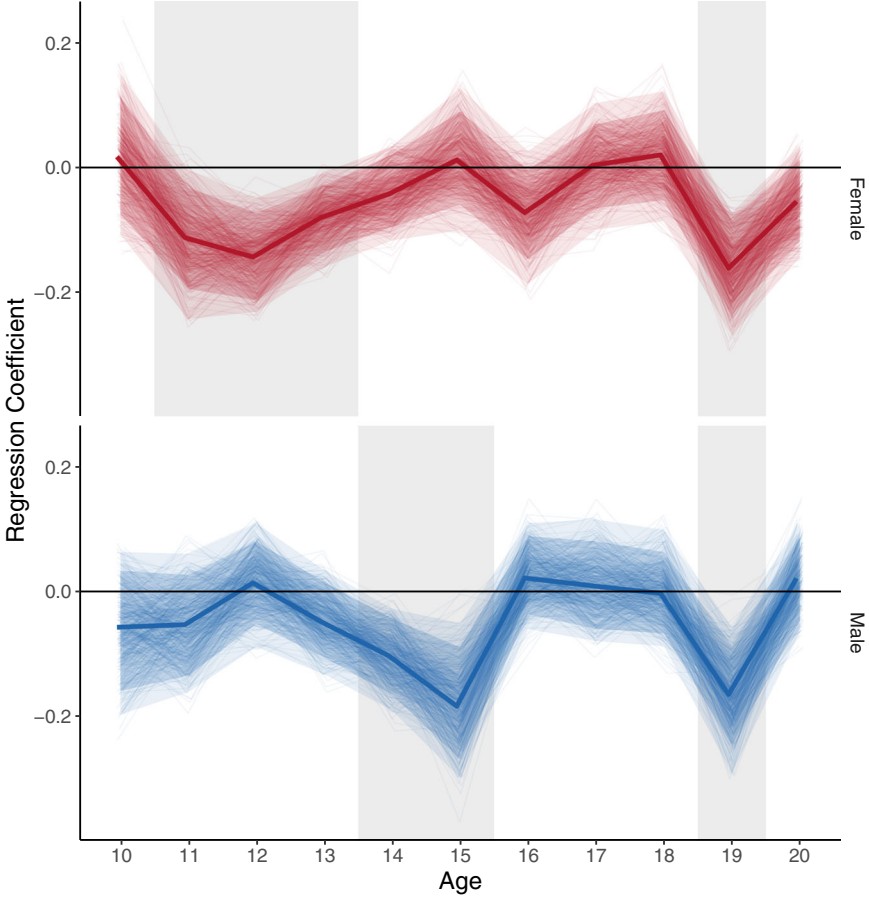

**Fig. 4 How social media use predicts life satisfaction in longitudinal data (ages 10–21).** Results from the cross-lagged path connecting estimated social media use to life satisfaction ratings one year later, estimated through a Random Intercept Cross-Lagged Panel Model of 17,409 participants (52,556 measurement occasions) aged 10–21. Results show how much an individual's deviation from their expected social media use at a certain age predicted a deviation from their expected life satisfaction ratings one year later (unstandardized estimates). Graph is split by sex (female = top/red, male = bottom/ blue) and the grey boxes indicate those ages where the path became statistically significant ($p < 0.05$, two-sided test). The thin lines represent the coefficients extracted from 500 bootstrapped versions of the model to visualize uncertainty, dark shaded ribbons represent bootstrapped 95% CIs, light shaded ribbons represent bootstrapped 99% CIs. The other cross-lagged path linking life satisfaction ratings to estimated social media use was constrained not to vary across age/sex and is not shown here. All tests are two-sided. Source data for this figure are provided as a Source Data file.

adolescent/adult survey. A two-sided Welch two-sample t-test for unequal variances comparing the scores of 15-year-olds on both younger adolescent and older adolescent/adult surveys showed no significance difference (Younger Adolescent $M = 5.64$, $N = 4095$; Older Adolescent/Adult $M = 5.25$, $N = 10$; $t(3) = 0.33$, $p = 0.77$, normality assumption not met, power to detect only a large difference of $d = 0.89$). A two-sided Bayesian two-sample t-test found anecdotal evidence in favour of the null hypothesis (i.e., no difference) between the two groups (BF10 = 0.49; variance of normal population: noninformative Jeffreys prior, standardized effect size: Cauchy prior; BF calculated via Gaussian quadrature). A second two-sided Welch two-sample t-test for unequal variances further found that 16-year-olds showed no significant difference in both surveys (Younger Adolescent $M = 5.86$, $N = 37$; Older Adolescent/Adult $M = 5.49$, $N = 4703$; $t(37) = 1.74$, $p = 0.09$, normality assumption not met, power to detect only a medium difference of $d = 0.46$), with a two-sided Bayesian two-sample t-test finding anecdotal evidence in favour of the null (BF10 = 0.51; variance of normal population: noninformative Jeffreys prior, standardized effect size: Cauchy prior; BF calculated via Gaussian quadrature). Further, we used linear regression on data of participants aged 13–18 years to predict life satisfaction ratings from both age and a categorical variable indicating the survey type (i.e., younger adolescent or older adolescent/ adult survey measurement). While age significantly predicted life satisfaction scores, the type of survey measurement did not (age: $b = -0.09$, se $= 0.01$, $p < 0.001$; survey type: $b = 0.05$, se $= 0.04$, $p = 0.140$; adjusted R-squared $= 0.016$, two-sided test; assumptions met except normality of residuals). With 24,533 participants included, this test would be highly sensitive (99% power) to extremely small effects ($f2 = 0.001$; an f2 value of 0.02 is small according to Cohen's guidelines) at a conventional alpha level of 0.05. A Bayesian approach comparing a regression including age and the survey category as predictors of life satisfaction scores against a regression only including age as a predictor, found the latter model to be a better fit for the data with a Bayes Factor of 11.5 ($+/- 1.4\%$; for priors see

Supplementary Methods 2). These analyses, therefore, support the conclusion that the measures are not qualitatively different.

We also created cross-sectional plots for additional well-being and mental health questionnaires completed only by the younger adolescent sample (age 10–15 years), these measures can be found in Supplementary Methods 1.

Social media use was measured at every wave for younger and older adolescent samples, and every three waves for adults. It entailed two questions in the younger adolescent survey: "Do you have a social media profile or account on any sites or apps?" (1 = Yes, 2 = No) and "How many hours do you spend chatting or interacting with friends through a social web-site or app like that on a normal school day?" (1 = None, 2 = Less than an hour, 3 = 1–3 h, 4 = 4–6 h, 5 = 7 or more hours). Prior to the final two waves, the questions were slightly different: "Do you belong to a social web-site such as Bebo, Facebook or MySpace?" and "How many hours do you spend chatting or interacting with friends through a social web-site like that on a normal school day?". For adults and older adolescent samples, the questions read: "Do you belong to any social networking web-sites?" (1 = Yes, 2 = No) and "How many hours do you spend chatting or interacting with friends through social web-sites on a normal week day, that is Monday to Friday?" (1 = None, 2 = Less than an hour, 3 = 1–3 h, 4 = 4–6 h, 5 = 7 or more hours). If a participant stated that they do not own a social media account or do not use social media to interact with friends, they were coded as the lowest score of 1; for the rest of the participants, we took their score on the second question, which measures how much time they spent interacting socially online.

Self-reported sex was reported annually ("male", "female"). When first surveyed, adults and older adolescents were asked to report their sex (options: "male" or "female"); subsequently, they were asked to confirm their sex collected in the previous waves. Younger adolescents were asked "are you male or female" (options: "male" or "female"); they were allowed to refuse response. If participants' report of sex varied between waves they were recorded as NA by the survey

administrators as part of a cumulative sex variable. Twenty-five measurement occasions reported sex as NA, which were too few to garner how their exclusion could have influenced our results. Due to the nature of this item design, we report the responses as "sex" in this manuscript; however, respondents may well have responded according to gender identity or gender assigned at birth based on genitalia as the nature of the questions was ambiguous, especially for younger adolescents. In the waves of data analyzed there was however no opportunity to examine to what extent self-report sex was related to gender identity.

Age for both the adolescent samples as well as the adult samples was derived by the data provider using self-reported date of birth and the interview date. Household income was measured using the log monthly total household net income for the household the adolescent belongs to (those households reporting 0 income were changed to 0.1 to allow for log transformation). The mean of all waves of available data pertaining to household income was taken to create the exogenous control variable used in the longitudinal models. Similarly, Index of Multiple Deprivation was derived by taking the Lower Layer Super Output Area (LSOA) of the participant and matching it to the governmentally derived rank on the Index of Multiple Deprivation (IMD; deciles). Again, the mean of IMD rank was taken to create a single exogenous control variable.

*Millennium cohort study.* For the Millennium Cohort Study, we analyzed a variety of mental health and well-being measures filled out by the adolescent or their primary caregiver respondent, a social media question completed by the adolescent respondent and demographic variables including sex and age. To measure social media use, adolescents were asked "On a normal week day during term time, how many hours do you spend on social networking or messaging sites or Apps on the internet such as Facebook, Twitter, and WhatsApp?" (1 = none, 2 = less than half an hour, 3 = half an hour to less than 1 h, 4 = 1 h to less than 2 h, 5 = 2 h to less than 3 h, 6 = 3 h to less than 5 h, 7 = 5 h to less than 7 h, 8 = 7 h or more). Sex and age were taken from the derived variable file of the MCS. The Millennium Cohort Study coded sex at birth: "male", "female" or "not known/not applicable", and age was calculated using age at last birthday.

Aligned with the Understanding Society well-being measures, adolescents were asked to fill out a well-being questionnaire on a scale from '1' (completely happy) to '7' (not happy at all) to indicate "How do you feel about the following parts of your life": your school work, the way you look, your family, your friends, the school you go to and your life as a whole. More extended well-being and mental health measured analyzed in the Supplementary can be found in the Supplementary Methods 1.

**Inclusion criteria.** In Understanding Society we excluded a variety of participants from the dataset. First, we excluded people aged over 80, because social media use was very low at higher ages (excluding 14,394 measurement occasions). We also excluded those 7 measurement occasions where a child aged 9 years filled out the younger adolescent survey. We further excluded those participants who completed a questionnaire twice in one age category (we only excluded the second time the questionnaire was filled out in one age category, excluding 9272 measurement occasions) and those whose sex was NA (25 measurement occasions; there were too few datapoints to treat this as a separate category). The latter could be due to them identifying as a different sex in the data collection frame or refusing to answer the question. These exclusions left a sample of 72,287 participants over 7 waves (1 wave = 12,444 participants, 2 waves = 8777 participants, 3 waves = 8794 participants, 4 waves = 7426 participants, 5 waves = 7099 participants, 6 waves = 9898 participants, 7 waves = 17,849 participants; see Supplementary Table 1 for measurement occasions by age and sex). The adolescent (10–21 years) sample used for longitudinal modelling consisted of 17,409 participants and 52,556 measurement occasions (see Supplementary Table 2 for measurement occasions by age and sex). In the Millennium Cohort Study, we excluded those not aged 13 or 14 (160 participants), leaving 11,724 participants (see Supplementary Table 3 for measurement occasions by age and sex).

There was dropout over time in the Understanding Society longitudinal adolescent sample (see Supplementary Fig. 9 and Supplementary Table 4). Due to the nature of our modelling approach, we were not able to integrate sampling weights into our estimation strategy. This limits the extent to which we can generalize our findings to the whole UK population.

**Cross-sectional analyses.** To examine the cross-sectional relations between estimated social media use and life satisfaction ratings across the life span we plotted life satisfaction scores by age and estimated social media use (Fig. 1, top). Furthermore, we plotted the amount of estimated social media use by age (Supplementary Fig. 1, middle), which showed how at very young and very old age ranges the limited number of high intensity social media users led to the substantial increases in error bars that make interpretation of trends at these ages unfeasible. To test whether the functional forms relating social media use estimates to life satisfaction ratings differ by sex across different ages we used an Akaike Weights procedure[36]. We used the r-package *lavaan* to fit different versions of the model (*life satisfaction ~ social media + social media²*, also including control variables) to the data: a multigroup model that freed both linear and quadratic terms between sexes, a model that freed only the linear or quadratic terms and a model that constrained both terms. While there was a spread between what model was

preferred in all ages, at ages 12, 13, and 14 the model constraining both the linear and quadratic term to be equal across sex was rejected (0% weight, Supplementary Fig. 10). When solely comparing a fully freed and fully constrained model we found that models that allowed sex variation in functional form were most favoured between 12 and 15 years (Fig. 1, bottom; Supplementary Fig. 3).

In the extension of the cross-sectional analyses, we analyzed a range of questionnaires that were only completed by 10–15-year-olds in the Understanding Society survey and questionnaires completed by 13- and 14-year-olds in the Millennium Cohort Study; additional mental health questionnaires were analyzed in Supplementary Methods 1 and Supplementary Results 1, while we also examined a variety of life satisfaction measures in the main manuscript. For the latter, we plotted each life satisfaction question's raw scores by social media use and age to examine whether a specific aspect of life satisfaction was more negatively related to estimated social media use (Fig. 2). We further used the Akaike weights procedure detailed above to examine the statistical evidence for sex differences.

**Longitudinal analyses.** To model the data longitudinally we used a Random Intercept Cross-Lagged Panel Model comparison framework[37], using the code structure provided by the Unified Framework of Longitudinal Models[40]. The model was selected due to its focus on within-person effects, without modelling general mean developments that have already been highlighted for both technology use and life satisfaction in adolescence[53,54]. The RI-CLPM model allowed us to focus on whether an individual's deviation from their expected level of a certain variable *y* (e.g., life satisfaction ratings) can be predicted from their prior deviation from their expected scores in another variable *x* (e.g., estimated social media use), while controlling for the structural change in *y* (e.g., life satisfaction ratings); and vice versa. We added two control variables, average log household income and the Index for Multiple Deprivation across all waves of data available for each participant, to account for the socioeconomic status of both the family and their immediate environment. We did not include time-varying control variables, such as income, at every wave or year of data collection because the model could not be fitted with the level of missingness present in the data.

We tested a variety of model constraints that force parameters to be equal across ages and sex, all of which were rejected: constraining the covariance of the residuals of latent factors after age 10 and constraining the residual variance of both social media and life satisfaction after age 10 ($\chi^2(63) = 2023$, $p < 0.001$); constraining the regression of the observed variables onto both mean IMD and mean log household income ($\chi^2(92) = 178$, $p < 0.001$); constraining the within-person carry-over effect for both social media estimates and life satisfaction ratings (also known as the within-person autoregression, positive carry-over means that a person who scores higher than their expected score is more likely to also score higher than their expected score in the following year; $\chi^2(42) = 218$, $p < 0.001$). A model constraining the cross-lagged paths of interest also suffered a significant drop in model fit: $\chi^2(42) = 77.7$, $p = 0.001$.

Having set up our core model, we first estimated a 'free model' where we allowed both cross-lagged paths (a deviation in social media use predicting a deviation in life satisfaction one year later; and vice versa) to vary across age and sex. The model was fit using robust Maximum Likelihood (MLR) to account for deviations from multivariate normality, and robust Huber-White Standard errors, and missing data were accounted for by Full-Information Maximum Likelihood (FIML) estimation[55]. All RI-CLPM parameter tests are two-sided Walds tests. We note here that FIML cannot guarantee to give unbiased estimates with missing exogenous variables, i.e. our control variables, as those are assumed to be measured without error. The model fit the data well: $\chi^2(434) = 1216.29$, $p < 0.001$, RMSEA = 0.014, [0.013, 0.015], CFI = 0.944, SRMR = 0.072. We extracted the value of the cross-lagged paths by age and sex, and plotted them in Fig. 3. We also fit this model to extended life satisfaction data collected for 10–15-year-olds and presented in Supplementary Fig. 7.

We then used model comparison and Akaike weights to examine whether models that constrained one or two of the cross-lagged paths (social media use predicting life satisfaction and life satisfaction predicting social media use) to be constant across age and sex fit better than the initial freed model. The model that constrained only social media use predicting social media use did not fit less well than a completely freed model ($\chi^2(21) = 27.8$, $p = 0.15$), while one that constrained only life satisfaction predicting social media use fit less well than the freed model ($\chi^2(21) = 49.5$, $p < 0.001$). The Akaike weights procedure showed that the model constraining only life satisfaction predicting social media use was more likely to be the best model for the data (99.2%), while the model freeing both (0.7%), the model constraining social media use predicting life satisfaction (0.0%) and the model constraining both (0.1%) were not (Supplementary Fig. 8).

Finding that the cross-lagged path from life satisfaction predicting social media use can be constrained across sex and age without loss of model fit, we, therefore, fit a second model with this constraint. The model fit was acceptable ($\chi^2(455) = 1,234.74$, $p < 0.001$, RMSEA = 0.01, [0.01, 0.02], CFI = 0.95, SRMR = 0.07). The model's carry-over effect paths of life satisfaction (average unstandardized b: male = 0.179, female = 0.229; average standardized β: male = 0.178, female = 0.220; average SE: male = 0.048, female = 0.044) and social media use (average unstandardized b: male = 0.327, female = 0.373; average standardized β: male = 0.315, female = 0.359; average SE: male = 0.039, female = 0.035) were predominantly positive, suggesting that individuals who scored higher on life

satisfaction or social media use than expected in one year were also more likely to score higher than expected on life satisfaction or social media use one year later. The cross-lagged paths are reported in the main paper. We further ran 500 bootstrapped samples of this model to examine the uncertainty around our estimates. We then plotted the value of the cross-lagged path of social media use predicting life satisfaction by age and sex in Fig. 4.

**Reporting summary**. Further information on research design is available in the Nature Research Reporting Summary linked to this article.

## Data availability

The Understanding Society and Millennium Cohort Study data used in this study have been deposited in the UK Data Service database. They are accessible after registration with the UK Data Service and completion of an End User Agreement (Understanding Society: University of Essex, Institute for Social and Economic Research, NatCen Social Research, Kantar Public. (2019). Understanding Society: Waves 1-9, 2009-2018 and Harmonized BHPS: Waves 1-18, 1991-2009. [data collection]. 12th Edition. UK Data Service. SN: 6614, Millennium Cohort Study: University of London, Institute of Education, Centre for Longitudinal Studies. (2020). Millennium Cohort Study: Sixth Survey, 2015. [data collection]. 6th Edition. UK Data Service. SN: 8156). Some of the data used in our study might not be accessible to users outside the UK. Please check the terms and conditions for each of the datasets prior to use on the UK Data Service. The Special License data necessary to calculate the Index of Multiple Deprivation in Understanding Society are available after an approval procedure through the UK Data Service. The figure and table data generated in this study are provided in the Source Data file. Amy Orben accessed all datasets.

Understanding Society is funded by the Economic and Social Research Council and various Government Departments, with scientific leadership by the Institute for Social and Economic Research, University of Essex, and survey delivery by NatCen Social Research and Kantar Public. The Millennium Cohort Study is funded by grants from the UK Economic and Social Research Council. Source data are provided with this paper.

## Code availability

The code for all analysis and data cleaning are available on the Open Science Framework: https://osf.io/fzspx/.

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

## Acknowledgements

This research was supported by a College Research Fellowship from Emmanuel College, University of Cambridge (A.O.), UK Medical Research Council MRC SWAG/076.G101400 (A.O.), the UK Economic and Social Research Council ES/T008709/1 (A.O. and A.K.P.), the Huo Family Foundation (A.K.P.), Wellcome Trust WT107496/Z/15/Z (S.J.B.), the Jacobs Foundation (S.J.B.), the Wellspring Foundation (S.J.B.) the University of Cambridge (S.J.B.) and the UK Medical Research Council SUAG/047 G101400 (R.A.K.).

## Author contributions

Conceptualization: A.O., A.K.P., S.J.B., and R.A.K. Data curation: A.O. Formal analysis: A.O. and R.A.K. Software: A.O. and R.A.K. Supervision: A.K.P., S.J.B., and R.A.K. Visualization: A.O. Writing - original draft preparation: A.O. Writing - review & editing: A.O., A.K.P., S.J.B., and R.A.K.

## Competing interests

The authors declare no competing interests.
