## [Peer Review File · Nature Communications]

Title: Windows of developmental sensitivity to social mediaREVIEWER COMMENTS

Reviewer #1 (Remarks to the Author):

This article examines two large datasets with information on both social media use and life satisfaction and reports that the association between social media use and lower life satisfaction largest in adolescence as compared to other age groups. The authors also report gender differences that are only present during the adolescent (12-15 years of age) window.

There is great deal to like about this paper in terms of the methodological rigor demonstrated in the longitudinal set of analyses, the clarity in the presentation of the results testing for stronger associations across age and gender, and the fit of the methods and analyses presented with questions related to adolescent wellbeing that are at the forefront of debates in research, education, and policy. The comments included below are intended to strengthen an already strong and well designed and executed set of analyses – with a strong recommendation that the methodological issues that arise in the first cross-sectional set of analyses would need to be addressed prior to publication.

First, the real strength of this paper is the within-person longitudinal set of analyses that document associations between life satisfaction and social media across the adolescent period (discussed below). However, the paper begins with the reporting of bivariate and cross-sectional correlations from the UK Understanding Society survey which includes study members spanning in age from 10 to 80 years of age. While the bi-variate associations represent an interesting starting point, this entire section relies on the presentation of unadjusted correlations between life satisfaction and social media.

- It was surprising that associations were not adjusted for factors that would be likely to a) confound the association between the two variables, and/or b) adjust for differences in the participants across age that may be otherwise driving associations. That is a is well documented that cohort effects may simply reflect the fact that the older participants cannot be treated as replicates of younger participants. This issue needs to be addressed to support the claim that social media has more robust associations with life satisfaction in the older versus younger ages (comparisons with adolescence of course are more easily justified, but are the 12 year-old and 60 year old individuals in the sample well matched?
- More generally, it would be useful to have information on how representative the sample of adolescents in this study are with respect to the larger population – there are comments embedded in the methods of the section referencing generalizability. However, additional assurances are required that this sample can be projected back onto the population and that adjustments have been made to account for differences that may exist in the sample across age.
- Given the fact that the second set of longitudinal analyses offer a much stronger and more focused set of tests of the main hypotheses, it raises the question of whether the bi-variate correlations presented in this first portion of the paper add significantly to this paper; one option may be to report them in the supplement or following the presentation of the longitudinal analyses given all of the threats to the

interpretation of these associations that arise due the questions about the sampling and the reliance on bi-variate correlations alone.

- Causal language in this section and throughout the manuscript should be avoided. That is, the term “sensitivity to social media” is used for bi-variate and unadjusted correlations between reported social media use and reported life satisfaction at a given time. There are no design features that place social media as the stimulus or cause of life satisfaction.

Second, application of the Longitudinal Random-Intercept Cross Lagged Panel Models across the adolescent samples generated a number of interesting findings. The analytic approach mapped well onto the main research questions and the results were clearly described and carefully documented and displayed. The availability of both the data and all script from the analysis represents a significant contribution. With that said, this section could be further strengthened by the following:

- The authors apply FIML to adjust analyses for missing data. This is a state-of-the art approach to addressing missing data. However, greater transparency is required with respect to the amount of missing data across waves and the degree to which selective attrition is likely to introduce bias (how does attrition vary across the key factors of family SES, social media use, life satisfaction, gender and other factors that may further bias the sample). Without these basic details it is difficult to determine how much of a threat missing data and any patterns of missingness threaten the interpretation of the coefficients.

- It was surprising that the work by Heffer and colleagues (<https://journals.sagepub.com/doi/10.1177/2167702618812727>) was not cited or integrated into the paper they have shown with longitudinal assessments of younger and older adolescence symptoms of depression predict social media use over time, but not vice versa, which was an effect that was found only in girls and was stronger in early versus later adolescence. The inclusion of this study would be important for two reasons. First, it is directly relevant to the question that is being addressed in the paper. Second, it provides evidence that depressive symptoms (closely related to life satisfaction) may be driving or influencing patterns of social media use over time, but not vice versa.

- Also, as described above, given that the longitudinal data are also observational causal language should be avoided throughout. E.g., “Figure 3: Longitudinal impact of social media use on life satisfaction”

This is an excellent paper which is likely to make a major contribution to the field.

Signed,

Candice Odgers, Professor of Psychological Science and Informatics, University of California Irvine

Reviewer #2 (Remarks to the Author):

Review for Nature Communications manuscript NCOMMS-21-01448-T, entitled: Windows of developmental sensitivity to social media.

This manuscript focuses on the relation between social media use on life satisfaction at different ages, using different datasets, and through both cross-sectional and longitudinal analyses. Overall I found the paper very well written, and easy to follow. I appreciate the thorough and thoughtful approach of the researchers, their careful explanations, and sophisticated analyses. I have two main suggestions how this already strong paper could be further improved.

First, it is clear that the interest here is in finding evidence for a causal relation between social media use and life satisfaction, and that the interest is not in just a statistical/predictive relation. This is apparent from terms and phrases like “windows of sensitivity”, “main driver”, “present use of social media has consequences for future life satisfaction”, “impact”, and “controlling for”. However, there could be readers who are offended by the idea of drawing conclusions about causality without having done an experiment. I therefore strongly recommend the paper by Miguel Hernán (2018; AJPH, <https://ajph.aphapublications.org/doi/10.2105/AJPH.2018.304337>), who provides a very appealing (and in my opinion) strong case for being more open and upfront about our interests in causality in non-experimental research, and how this could be done. Specifically, he writes: “Therefore, the term “causal effect” is appropriate in the title and Introduction section of our article when describing our aim, in the Methods section when describing which causal effect we are trying to estimate through an association measure, and in the Discussion section when providing arguments for and against the causal interpretation of our association measure. The only part of the article in which the term “causal effect” has no place is the Results section, which should present the findings without trying to interpret them.” To be clear, I am not suggesting the paper should be totally rewritten, or that every other sentence should contain the term “causal”, but I do think the point that the interest is in uncovering an underlying causal mechanism, could be made a bit more clearly in the manuscript, for instance in the introduction. Of course, more careful formulations such as “evidence for a causal effect” could also be used (as for instance is *almost* done on p. 11: “[...] provides evidence for the hypothesis that the impact of social media on individuals varies [...]”)

Second, with the longitudinal analyses, it is very important to point out that lagged relations (both autoregressive and cross-regressive) are a function of the time-interval between the observations. Hence, if one does not find a cross-lagged relation from X to Y, this should not be taken to mean that X does not have an effect on Y; rather, it implies that it does not have an effect over this time-interval. This point has been raised by many over the years (e.g., by Gollob and Reichardt, 1987; Child Development, https://www.jstor.org/stable/1130293?seq=11#metadata_info_tab_contents), and more recently has been discussed by various researchers who advocate a continuous time modeling approach. For instance, Dorman and Griffin (2015; Psychological Methods, <https://pubmed.ncbi.nlm.nih.gov/26322999/>), very clearly describe this issue in the context of cross-lagged models, and argue that many of our cross-lagged panel studies are based on time intervals that

are too long to properly capture the dynamics of the actual underlying process. Specifically, they state: “Our call for researchers to use shorter time lags implies that the common lag of 1 year should be supplemented with shortitudinal studies of much shorter time lags.” When considering social media use and life satisfaction, one could raise the question whether an interval of 1 year is not too long to see any within-person dynamics. Of course, this is a restriction the authors have to work with (as the data were gathered in this way), but some discussion of the (likely) possibility that other results would be found if the interval was 1 month instead of a year, or just 1 week, 1 day, or 1 hour, is important to paint the bigger picture of this kind of research.

Minor issues:

- a) I found Figure 1 rather confusing at first. I think this is because in the description it says “Cross-sectional correlation between estimated social media use and a one-item satisfaction with life measure”; this made me expect to see correlations plotted against something else (i.e., correlation on the y-axis and something else on the x-axis). Instead, what is plotted is the average life satisfaction score for different degrees of social media use at different ages. I think that this could be stated more clearly.
- b) From the description of Figure 3 (which I btw think is a very smart and appealing way to represent these results!), it is not clear whether these “cross-lagged path linking estimated social media use to life satisfaction” is from media use to satisfaction or from satisfaction to social media use; probably the former, but it could be stated more clearly (e.g., saying it is the cross-lagged path from estimated social media use towards life satisfaction one year later).
- c) I am wondering whether how (or if) the inverted U-shaped relations discussed in the context of Figure 1 relate to the results in the longitudinal analyses.
- d) When discussing the comparison between the different life satisfaction measurements at different ages (on p. 19), the authors indicate there was a small inconsistency in the data collection, providing them with 37 cases on which they can test whether there are differences. While I appreciate this point, I think it is also important to indicate that the power to detect differences is not that large with 37 cases (I assume).
- e) On p. 19 it is also indicated that additional variables were included in the analyses (more extensive measures of life satisfaction, self esteem, SDQ. It is not clear to me what was done with these variables exactly; were they also analyzed using a RI-CLPM, or where they included in the main RI-CLPM analysis?
- f) On p. 21 it is stated that there were two variables included as exogenous control variables (time-invariant). Looking at the lavaan code I see these were included as direct predictors of the observed variables, and that their effect was allowed to vary over time (i.e., regression coefficients were not invariant over time). This could be explained in the text. Note that Mulder and Hamaker recently published a paper with extensions of the RI-CLPM, which includes a discussion of how to include such time-invariant covariates, and what the effect of different ways of including them are (see <https://www.tandfonline.com/doi/full/10.1080/10705511.2020.1784738>).
- g) On p. 27 the authors discuss certain constraints over time that they tried out; they refer to covariances and variances after age 10; note that in the model you estimate the residual variances and covariances between residuals, when variables are endogenous (dependent); constraining residual variances and covariance does not guarantee that the variances of the (within-person) variables will be invariant over time (this depends on the lagged parameters, and also on the initial variances and

covariance).

h) On p. 28, the chi-square difference test is discussed for constraining all the cross-lagged paths from life satisfaction to social media use. This test has 21 df, which would mean that it has 21 parameters less than the model without these constraints. Is that correct? I am confused where all these df come from.

i) The authors use the term carry-over to refer to the autoregression. Although this is a useful term, I think that it would also be helpful for some readers to indicate this is the same as the (within-person) autoregression.

Ellen Hamaker

Reviewer #3 (Remarks to the Author):

This is a comprehensive and fascinating look at the data on social media use and life satisfaction across age groups. In my view, the paper would be strengthened by some minor revisions to the abstract and text. With the two most important points denoted by asterisks, these include:

1. p. 4: The paper states: "Research has already highlighted gender differences in the links between social media use and well-being in adolescence, although these links have repeatedly been labelled as negligible."

This is a strange statement, as several studies examining links between social media use and well-being have found substantial differences, especially among girls (e.g., twice as many heavy users depressed vs. non-users). These studies include:

Kelly, Y., Zilanawala, A., Booker, C., & Sacker, A. (2018). Social media use and adolescent mental health: Findings from the UK Millennium Cohort Study. *EClinicalMedicine*, 6, 59-68.

Twenge, J. M., & Martin, G. N. (2020). Gender differences in associations between digital media use and psychological well-being: Evidence from three large datasets. *Journal of Adolescence*, 79, 91-102.

Viner, R. M., Aswathikutty-Gireesh, A., Stiglic, N., Hudson, L. D., Goddings, A-L., Ward, J. L., & Nicholls, D. E. (2019). Roles of cyberbullying, sleep, and physical activity in mediating the effects of social media use on mental health and wellbeing among young people in England: a secondary analysis of longitudinal data. *Lancet Child and Adolescent Health*.

These papers should likely be cited, and the phrase about the links being labeled as negligible should be deleted.

2. p. 5:

-- How did the authors decide to label 12-15 years as early adolescence and 16-21 years as later

adolescence? No citation is given. Or was this division done post-hoc after the results were known?
-- This division is also confusing as the introduction defines adolescence as beginning at age 10, yet here it begins at age 12; the introduction also defines adolescence as lasting until age 24, yet here it ends at age 21.

3. p. 6: "This refutes claims that social media displays solely negative relations with well-being ..." No citation is given for this statement. Who has made these claims?

4. p. 7 "Younger adolescents showed a different relation than older adolescents: it was less curvilinear and also showed more prominent gender differences in its form." Describing this association as "less curvilinear" is odd – especially for girls, the association shown in Figure 1 is almost completely linear at ages 12, 13, and 14. Given that, it should probably be described for what it is: linear, not "less curvilinear."

5. In Figure 2A, what is "factor score" on the y-axis? Is this a Z-score or standardized score, or something else? If it's a Z-score, how were the data standardized? (within study? Within gender?)

**6. Another recent paper used the same dataset and came to similar conclusions, yet that paper is not cited here. The authors should say explicitly how their paper is different from the previous research: Booker, C. L., Kelly, Y. J., & Sacker, A. (2018). Gender differences in the associations between age trends of social media interaction and well-being among 10-15 year olds in the UK. *BMC Public Health*, 18(1), 321.

7. It is unclear why the authors cite reference 5 (Ophir) as an example of "widespread concern" about social media, as this paper was not about concern around social media. Instead, it was focused on a critique of another article – a critique the authors of the original article refuted in their reply. Thus, it seems like an odd choice for a citation.

**8. The abstract notes that "Decreases in life satisfaction predicted small subsequent increases in social media use." That suggests life satisfaction changes before social media use. However, the explanation below Figure 3 notes that for certain ages, the link goes the other way: "an individual's increase from expected levels of estimated social media use predicted a decrease from their expected levels of life satisfaction one year later" – meaning that social media use changes before life satisfaction. The text echoes the figure (p. 9, lines 8-10). The abstract implies that the association is unidirectional (with life satisfaction coming before social media use), but the text and figure show that it is bidirectional, especially at certain ages. The abstract should be brought in line to agree with the text and figure.

Reviewer #4 (Remarks to the Author):

This study investigates associations between social media use and life satisfaction in a large sample in

the UK. A focus is on age and gender differences, and directional associations are tested in the youth sample. More social media use is associated with lower life satisfaction (although some quadratic effects are observed). Gender effects are found in adolescents but not adults. Bidirectional associations are found in youth, with some evidence that social media use predicts later life satisfaction during specific age periods in adolescent males and females. I appreciate the aims of the analyses, the sample is large, and the authors have done a really nice job with story-telling. However, in my view the authors have not done enough here to test their aims or go beyond what has already been shown in the literature. A number of statements are not supported by statistical tests. Effect sizes are not presented. The directional nature of associations are most interesting, although it is difficult to know what to conclude here – analytic techniques such as causal mediation could be employed to better address questions. Other specific comments below.

Background: The authors use the term gender when I believe they mean “sex” (i.e., when referring to bodily changes in male and female adolescents). The authors state that participants were asked about “sex”. Then why use “gender” throughout?

Figure 1: Nonlinear associations between social media use and life satisfaction appear to be present in a number of age groups (not just adolescents). Is this the case?

Figure 1: Do associations hold after controlling for socioeconomic status?

The life satisfaction question for youth vs adults was different, with the youth question asking about happiness (rather than satisfaction). The authors try to address this by comparing small samples of 15 and 16 year olds who received the wrong survey. 15 and 16 year olds do not differ that much in age, so the meaningfulness of this is highly questionable. Therefore, different measures could be a reason for different patterns across the lifespan and should be acknowledged.

Where are the statistical tests performed to support claims of sex/gender differences (Figure 2)?

The authors plot associations between social media use and different domains of life satisfaction and mental health (in Figure 2). Then it appears that the authors only eyeball these plots to support claims about gender differences. This seems a missed opportunity to look more specifically at the association between social media use, life satisfaction and mental health. Was social media use statistically differentially related to different domains of life satisfaction? Can poor mental health potentially explain associations?

For question 2, why did the authors control for mean household income, as opposed to income at each wave?

The authors state that “The gender difference in timing suggested that early increases in sensitivity to social media might be due to maturational processes such as puberty, which occur earlier in girls compared to boys”. This is overly speculative given that puberty was not investigated.

Were statistical tests performed to support claims of gender differences in Figure 3?

Were the prospective effects of social media use on life satisfaction weaker/stronger than those of life satisfaction on social media use?

Please state effect sizes for all findings.

It is stated on one section that latent factors were measured for the SDQ, but in another section is stated that the emotion and conduct subscale scores were used. Please clarify.

Reviewer #5 (Remarks to the Author):

Manuscript number: NCOMMS-21-01448-T

Title: Windows of developmental sensitivity to social media

This manuscript examined the evidence on the relationship between social media and life satisfaction during adolescence using data from the UKHLS and Millennium Cohort Study. As the authors comment, this is a much disputed field and new evidence is always welcome. It extends previous work on gender differences by adding a longitudinal perspective and a different outcome, life satisfaction. The authors conclude that social media use predicts a small decrease in life satisfaction one year later, occurring at different ages depending on gender, and that decreases in life satisfaction also predicted small subsequent increases in social media use.

The abstract is quite confusing as it first mentions an age range of 10-80 years. Then reports that that the cross-sectional relationship between estimated social media use and life satisfaction was more negative in adolescence (12-15 years), compared to other age groups. Then longitudinal analyses are based on 10-21 year olds. Why all these different age ranges?

From figure 1, it looks to me as though estimated social media use and life satisfaction was more negative at age 11 too. I could find no mention on how the conclusion that the correlation was more negative at ages 12-15 years was made, only that there were tests for gender differences at different ages. Secondly, in what way did social media use predicted a decrease in life satisfaction? Was it higher or lower SMU?

The differences in the measurement of life satisfaction in the youth and adult surveys raise many questions. I am not convinced by the Welch two-sample t-tests when there are such large differences in sample sizes for the two groups. The items on overall life satisfaction come after other individual items

which vary between the youth and adult surveys and must differentially prime the respondents for the overall question.

Why were 12-15 year olds and 16-21 year olds compared? Justification would be helpful since the UN define adolescence as the period between 10 and 19 years of age. The WHO also defines adolescents as individuals aged 10-19 years. So why drop the 10-11 year olds in UKHLS and extend adolescence to 21? A more hypothesis driven analysis would have usefully dispelled thoughts of cherry-picking results.

Figure 1 suggests that the age-period-cohort problem could be present in the data. For example, mean SMU/LS correlation at an 18 year old girl's 1st observation is ~5, but mean SMU/LS correlation at a 13 year old girl's last observation is more like 4.5. Yet there is no adjustment for birth year or wave in the longitudinal analyses.

If you refer to male/female, then you are really talking about sex not gender. If you are using the term gender to mean that you are interested in the socially constructed influences on social media behaviour and life satisfaction, then it would be best to use the terms men/women throughout irrespective of the labels used in the original surveys. Alternatively, using male/female terminology infers that you believe that biological influences are at play despite standing on the fence with "biological, cognitive or social" (page 4, line 14). Which is it?

I did not understand the model selection strategy. First, I did not notice any mention of the estimation method. I am less familiar with the R SEM implementation than other statistical packages. Is the chi-square test appropriately estimated, depending on the estimation method? This seems important, as the BIC statistics favour simpler measurement models than those selected by the authors based on the chi-sq difference test. Second, the authors move on to use AIC weights for the comparison of the longitudinal models. Would the same conclusions have been drawn if BIC weights were used instead?

There was no discussion of the limitations of the study. I believe this is necessary as this is an observational study and there appears to be some subjectivity in the methods employed (see below).

There is also no discussion of how these new findings fit with or contradict findings from previous work on the relationship between SMU and adolescent wellbeing. Do the longitudinal findings confirm or refute findings from cross-sectional analyses? Do relationships with life satisfaction agree with other dimensions of wellbeing such as depression? Are the supplementary analyses using internalising and externalising symptoms congruent with previous research with these outcomes?

Other points

More important that the "72,281 10-80 year-old participants" (page 5, line 9), is information on the number of younger and older adolescents, and the average number of observations.

Can you confirm the age range for "late adolescence" (page 6, line 20).

Discussing the model constraints on page 10, “A model constraining the path of life satisfaction predicting estimated social media use to be constant across age and gender, while allowing the path of estimated social media use predicting life satisfaction to vary, was highly preferred over other model constraints”. Surely more to the point, is whether the constrained model fits as well as an unconstrained model?

Could it also be that case that higher LS predicted a negative change in SMU rather than a lower than expected LS score in one year predicted a positive change in SMU (page 10, line 24)? See also the abstract: in what way did social media use predicted a decrease in life satisfaction? Was it higher or lower SMU?

Can it be clarified what the age range is for “young adults” (page 20, line 8). It is slightly confusing as the methods text suggests this is 16-21 (page 18, line 10) but figure 1 shows detail for ages 20-25 and 26-29 and 10 year bands thereafter. What is special about the 20s?

When age was defined (page 21, line 9), did the authors use the interview date or the sampling wave/year?

Why exclude the 15 and 16 year olds who filled out the incorrect survey when you have previously reported that it did not make any difference whether the youth or adult questionnaire was used?

I presume that participants who completed a questionnaire twice in one age category must have had a late interview in one wave and an on-time interview in the next wave. If so, this addresses my previous question about defining age. But if age is calculated using year/month information and then rounded to the nearest year, then 9k respondents would not have to be dropped.

If sampling weights were not applied (why?) then would it not have been possible to have at least adjusted all models using the sampling strata variables?

04 June 2021

Response to Reviewers

We thank the reviewers for the very insightful and important points raised during the initial review process. Throughout this document we detail the changes adopted in the manuscript to address them.

Reviewer 1

This article examines two large datasets with information on both social media use and life satisfaction and reports that the association between social media use and lower life satisfaction largest in adolescence as compared to other age groups. The authors also report gender differences that are only present during the adolescent (12-15 years of age) window.

There is great deal to like about this paper in terms of the methodological rigor demonstrated in the longitudinal set of analyses, the clarity in the presentation of the results testing for stronger associations across age and gender, and the fit of the methods and analyses presented with questions related to adolescent wellbeing that are at the forefront of debates in research, education, and policy. The comments included below are intended to strengthen an already strong and well designed and executed set of analyses – with a strong recommendation that the methodological issues that arise in the first cross-sectional set of analyses would need to be addressed prior to publication.

Reviewer 1, Point 1: *First, the real strength of this paper is the within-person longitudinal set of analyses that document associations between life satisfaction and social media across the adolescent period (discussed below). However, the paper begins with the reporting of bivariate and cross-sectional correlations from the UK Understanding Society survey which includes study members spanning in age from 10 to 80 years of age. While the bi-variate associations represent an interesting starting point, this entire section relies on the presentation of unadjusted correlations between life satisfaction and social media.*

It was surprising that associations were not adjusted for factors that would be likely to a) confound the association between the two variables, and/or b) adjust for differences in the participants across age that may be otherwise driving associations. That is a is well documented that cohort effects may simply reflect the fact that the older participants cannot be treated as replicates of younger participants. This issue needs to be addressed to support the claim that social media has more robust associations with life satisfaction in the older versus younger ages (comparisons with adolescence of course are more easily justified, but are the 12 year-old and 60 year old individuals in the sample well matched?).

To address this comment we made a variety to changes that allow us to take into account necessary controls in our two main cross-sectional analyses.

Firstly, we now replicate our cross-sectional findings displayed in the raw data plot (Figure 1) in analyses that allow us to add the control variables of household income, neighbourhood deprivation (Index of Multiple Deprivation) and year of data collection. Specifically, we run a regression for each

age group where social media use and the control variables predict life satisfaction. We then plot the regression coefficient linking social media use to life satisfaction by age in Extended Data Figure 2 and reference it in our main text:

“Furthermore the finding was robust when adding control variables of income, neighbourhood deprivation (measured using the Index of Multiple deprivation) and year of data collection: adolescence was the only time the cross-sectional regression coefficient of social media use predicting life satisfaction fell below -0.1 units (Extended Data Figure 2).” (Lines 111-115)

Extended Data Figure 2: Cross-sectional relation between social media use and life satisfaction across age

The graph plots the regression coefficient predicting life satisfaction from social media use (y axis) for different age groups (x axis). The confidence intervals represent the standard error of the regression coefficient. The grey rectangle highlights the area of the graph where the regression coefficient goes below ± 0.1 units. The regression includes the covariates of household income, neighbourhood deprivation and year of data collection. It shows that the link between social media use and life satisfaction are most negative in early adolescence for females rather than males.

Secondly, we now add the control variables (household income, neighbourhood deprivation and year of data collection) to our analyses of sex differences in the cross-sectional data. Specifically we add them to our regressions using the AIC weights:

“This difference between males and females is statistically supported by an Akaike weights procedure³⁶ showing that models associating estimated social media use and life satisfaction while differentiating for self-reported sex (and controlling for household income, neighbourhood deprivation and year of data collection) were preferred between the ages of 12 and 15 (AIC weights ratios ranging from 5:1 to over 6,000,000:1 in favour of the model differentiating males and females).

In contrast, models differentiating for sex were not preferred at other ages (Figure 1, bottom; Extended Data Figure 3).” (Lines 151-157)

These analyses including the control variables are now displayed at the bottom of Figure 1 and in the Extended Data.

Reviewer 1, Point 2: *More generally, it would be useful to have information on how representative the sample of adolescents in this study are with respect to the larger population – there are comments embedded in the methods of the section referencing generalizability. However, additional assurances are required that this sample can be projected back onto the population and that adjustments have been made to account for differences that may exist in the sample across age.*

We thank the reviewer for raising this question. The Understanding Society sample was “designed to be representative of the UK population” (Lynn, 2009) while the Millennium Cohort Study “is a birth cohort study of a sample of around 11,000 young people born between September 2000 and January 2001. The study over-sampled some parts of the population, for example children from ethnic minorities or those living in disadvantaged areas; there is no sample refreshment by immigrants”. While sampling weights would have been a welcome addition to the manuscript, to make sure the samples studied in this manuscript were representative, the sampling weights provided by Understanding Society were given per year (i.e. sampling weight for 2018 for example). Due to the innovative and unusual nature of our data analysis (examining data by age rather than by year), the sampling weights do not correspond to our analytical design. We therefore instead noted the lack of sampling weights as a limitation in our study:

“Due to the nature of our modelling approach we were not able to integrate sampling weights into our estimation strategy. This limits the extent to which we can generalize our findings to the whole UK population.” (Lines 608-610)

Each adolescent sampled is a member of one of the households surveyed. As some households however drop out more rapidly over time, and this could lead to bias, Understanding Society has added booster samples (e.g. for those of ethnic minority background) to ensure that households in the study remain representative over time. This should ensure that adolescents are comparable, at least in some important respects, to the adult sample in the study.

Reviewer 1, Point 3: *Given the fact that the second set of longitudinal analyses offer a much stronger and more focused set of tests of the main hypotheses, it raises the question of whether the bi-variate correlations presented in this first portion of the paper add significantly to this paper; one option may be to report them in the supplement or following the presentation of the longitudinal analyses given all of the threats to the interpretation of these associations that arise due the questions about the sampling and the reliance on bi-variate correlations alone.*

We appreciate the reviewer’s comment about the utility of the cross-sectional analyses and seriously considered moving these analyses into the supplementary materials. Yet the cross-sectional analyses form a core part of the manuscript’s analytical reasoning, as they highlight the need to examine adolescence more closely in subsequent analyses. Further they highlight the importance of examining sex differences in our modelling. We therefore feel like moving them completely to the supplementary materials would not provide an accurate overview of our scientific process.

However, we have moved parts of the cross-sectional analyses (parts of Figure 2 and related text) to the supplementary materials. Further, we have made some changes to make the framing of the cross-sectional analyses clearer throughout the manuscript, e.g. in the introduction:

“To investigate the existence of developmental windows of sensitivity to social media, we first use cross-sectional data to examine whether adolescence might represent a period during which the association between well-being and social media is different in comparison to other life stages, and if differences between males and females are present during this time. Using longitudinal data, we then test the idea that there exist sex-specific windows of sensitivity to social media during adolescence itself.” (lines 96-101)

Reviewer 1, Point 4: *Causal language in this section and throughout the manuscript should be avoided. That is, the term “sensitivity to social media” is used for bi-variate and unadjusted correlations between reported social media use and reported life satisfaction at a given time. There are no design features that place social media as the stimulus or cause of life satisfaction.*

We agree with the reviewer that causality is both an interesting and challenging aspect of this study, as it was observational in nature. The issue of causality was also raised by Reviewer 2 in their Point 1. Yet Reviewer 2 provided contradictory recommendations that we should make causal claims more explicit, but also clearly cover the assumptions necessary for them to be supported. We have reflected on this feedback extensively while also consulting the relevant literature laying out best practices with regard to causality. As Reviewer 1 recommends, we have removed all causal language from the cross-sectional part of the study. For example, we have done so in the section of the manuscript highlighted in Reviewer 1, Point 3, and in the figure legends:

“Figure 1: Social media use and life satisfaction across the lifespan

Top: The cross-sectional relationship between social media use and a one-item life satisfaction measure for 72,281 UK participants between the age of 10 and 80 years.”

“Figure 3: How social media use predicts life satisfaction in longitudinal data (ages 10-21)”

Further, for the longitudinal part of the manuscript we removed most causal language, while retaining the phrasing of “sensitivity to...” as this is important for the core inferences of the manuscript. We have added a paragraph making explicit the assumptions that would need to be met for our longitudinal analyses to truly represent causal inferences.

“The study has multiple limitations that need to be considered. First, to draw estimates of causal effects from these analyses one would need to adopt the following assumptions: a) there are no time-varying covariates that impact the relation between social media use and life satisfaction; b) the time interval between studies (one year) is the right length to capture the effects of interest; and c) the bidirectional links estimated by our longitudinal model are linear in nature.” (Lines 297-302)

Reviewer 1, Point 5: *Second, application of the Longitudinal Random-Intercept Cross Lagged Panel Models across the adolescent samples generated a number of interesting findings. The analytic approach mapped well onto the main research questions and the results were clearly described and carefully documented and displayed. The availability of both the data and all script from the analysis represents a significant contribution. With that said, this section could*

be further strengthened by the following:

The authors apply FIML to adjust analyses for missing data. This is a state-of-the art approach to addressing missing data. However, greater transparency is required with respect to the amount of missing data across waves and the degree to which selective attrition is likely to introduce bias (how does attrition vary across the key factors of family SES, social media use, life satisfaction, gender and other factors that may further bias the sample). Without these basic details it is difficult to determine how much of a threat missing data and any patterns of missingness threaten the interpretation of the coefficients

We understand the reviewer's concern about the threat of missing data, and patterns of missingness that might threaten our interpretation of analyses. We now report the amount of missing data per age group to show that we have sufficient power to do our analyses in the supplementary materials:

Age	Number of Measurement Occasions
10	3895
11	4048
12	4149
13	4229
14	4272
15	4105
16	4740
17	4889
18	4804
19	4641
20	4435
21	4349

Extended Data Table 1: Number of measurement occasions by age in younger and older adolescents in Understanding Society (ages 10-21)

Because the data is a rolling sample, with participants aging in and out of certain surveys at different times, and with data split by age rather than wave of data collection, it is very difficult to give a comprehensive overview of dropout. However, to provide an assessment of the risk of selective attrition we examined five age cohorts of participants who remain part of the adolescent sample for the whole of the study (those aged 10, 11, 12, 13, 14 and 15 at wave 1). These analyses can now be found in the extended data section of the manuscript. We first examined the amount of missing data

found across ages as the participants took part in the subsequent waves, showing that there is dropout over time and this is more pronounced at higher ages:

“Extended Data Figure 9: Missing Data (%) by Age

To examine how missingness develops over both younger and older adolescent cohorts we investigated the percentage missingness in those age cohorts who remained in the adolescent survey for all seven waves of data collection (those aged 10, 11, 12, 13, 14 and 15 during the first wave). We find that missingness increases over time, but that it increases faster in later age cohorts. This shows that attrition is greater at older ages.”

Next we predicted whether those surveyed at wave 1 would still be surveyed at wave 7, using sex, income, index for multiple deprivation, social media use at the first wave and life satisfaction at the first wave as predictors. We found that missingness was consistently predicted by sex (higher likelihood of dropout if male) and income (higher likelihood of dropout if low income). As both these measures are either intrinsic to the analysis (e.g. in the multigroup model) or used as a covariate (e.g. IMD) this suggests that our FIML should be able to accommodate this well. There were no other clear trends in terms of selective dropout over time. This is now also described in the extended data section:

Predictors	missing_10			missing_11			missing_12			missing_13			missing_14			missing_15		
	Odds Ratios	CI	p	Odds Ratios	CI	p	Odds Ratios	CI	p	Odds Ratios	CI	p	Odds Ratios	CI	p	Odds Ratios	CI	p
(Intercept)	1.94	0.12 – 32.21	0.641	0.03	0.00 – 0.52	0.018	0.03	0.00 – 0.32	0.005	0.18	0.01 – 2.02	0.171	0.02	0.00 – 0.20	0.002	2.12	0.21 – 20.98	0.520
ls	0.86	0.75 – 0.98	0.023	1.04	0.91 – 1.20	0.570	1.09	0.96 – 1.25	0.182	0.96	0.84 – 1.10	0.582	1.05	0.92 – 1.20	0.488	0.99	0.86 – 1.15	0.904
sm	0.81	0.64 – 1.01	0.064	1.04	0.86 – 1.26	0.673	0.86	0.73 – 1.02	0.089	0.91	0.77 – 1.07	0.253	0.84	0.72 – 0.98	0.028	0.85	0.73 – 1.00	0.051
imd	1.05	0.99 – 1.11	0.077	1.01	0.96 – 1.07	0.637	1.00	0.94 – 1.05	0.854	1.01	0.95 – 1.07	0.751	1.04	0.99 – 1.10	0.138	1.03	0.97 – 1.09	0.310
inc	1.04	0.74 – 1.46	0.831	1.55	1.08 – 2.26	0.019	1.55	1.14 – 2.14	0.006	1.29	0.96 – 1.76	0.095	1.60	1.17 – 2.21	0.003	0.92	0.70 – 1.21	0.536
sex [male]	0.74	0.54 – 1.00	0.047	0.67	0.49 – 0.91	0.010	0.62	0.46 – 0.83	0.001	0.71	0.52 – 0.96	0.026	0.78	0.58 – 1.05	0.107	0.57	0.42 – 0.78	<0.001
Observations	700			681			764			727			768			724		
R ² Tjur	0.022			0.020			0.028			0.013			0.032			0.022		

“Extended Data Table 2: Examining Selective Attrition

In addition to the analyses in Extended Data Figure 9, we examined selective attrition further in those age cohorts who remained in the Understanding Society adolescent survey for all seven waves of data collection (those aged 10, 11, 12, 13, 14 and 15 during the first wave). To do so, we coded those adolescents as ‘1’ who had data for wave 1 and wave 7, and those as ‘0’ for those who had data for only wave 1 and not wave 7. We then regressed this attrition variable onto the predictor variables of sex, mean income (inc), mean index of multiple deprivation (imd), social media use at wave 1 (sm) and life satisfaction at wave 1 (ls). We found that the only consistent predictor of attrition is sex (males showing higher levels of attrition). Further, for half of the age cohorts lower income predicts higher levels of attrition.”

Reviewer 1, Point 6: *It was surprising that the work by Heffer and colleagues was not cited or integrated into the paper they have shown with longitudinal assessments of younger and older adolescence symptoms of depression predict social media use over time, but not vice versa, which was an effect that was found only in girls and was stronger in early versus later adolescence. The inclusion of this study would be important for two reasons. First, it is directly relevant to*

the question that is being addressed in the paper. Second, it provides evidence that depressive symptoms (closely related to life satisfaction) may be driving or influencing patterns of social media use over time, but not vice versa.

We thank the reviewer for the suggestion and have now added both Heffner et al. and Jensen et al. to the introduction:

“Yet, even after years of research, there is still considerable uncertainty about how social media use relates to well-being. Meta-analyses have located small or negligible negative links between social media use and well-being^{6,7}, while experimental evidence is mixed^{8,9}. Longitudinal observational studies that have investigated the predictive relationships between social media use and well-being have found that they are either reciprocal¹⁰, only present in a certain direction or sex¹¹ or not present at all¹². The lack of concrete evidence is an issue routinely highlighted by academics³, medical professionals^{13,14} and policymakers^{15,16}.” (Lines 46-53)

Reviewer 1, Point 7: *Also, as described above, given that the longitudinal data are also observational causal language should be avoided throughout. E.g., “Figure 3: Longitudinal impact of social media use on life satisfaction”*

We have addressed this and changed the figure legend. Please see our response to Reviewer 1 Point 3 for further details.

“Figure 3: How social media use predicts life satisfaction in longitudinal data (ages 10-21)”

Reviewer 2

This manuscript focuses on the relation between social media use on life satisfaction at different ages, using different datasets, and through both cross-sectional and longitudinal analyses. Overall I found the paper very well written, and easy to follow. I appreciate the thorough and thoughtful approach of the researchers, their careful explanations, and sophisticated analyses. I have two main suggestions how this already strong paper could be further improved.

Reviewer 2, Point 1: *First, it is clear that the interest here is in finding evidence for a causal relation between social media use and life satisfaction, and that the interest is not in just a statistical/predictive relation. This is apparent from terms and phrases like “windows of sensitivity”, “main driver”, “present use of social media has consequences for future life satisfaction”, “impact”, and “controlling for”. However, there could be readers who are offended by the idea of drawing conclusions about causality without having done an experiment. I therefore strongly recommend the paper by Miguel Hernán (2018; AJPH, <https://ajph.apbapublications.org/doi/10.2105/AJPH.2018.304337>), who provides a very appealing (and in my opinion) strong case for being more open and upfront about our interests in causality in non-experimental research, and how this could be done. Specifically, he writes: “Therefore, the term “causal effect” is appropriate in the title and Introduction section of our article when describing our aim, in the Methods section when describing which causal effect we are trying to estimate through an association measure, and in the Discussion section when providing arguments for and against the causal interpretation of our association measure. The only part of the article in which the term “causal effect” has no place is the Results section, which should present the findings without trying to interpret them.” To be clear, I am not suggesting the paper should be totally rewritten, or that every other sentence should contain the term “causal”, but I do think the point that the interest is in uncovering an underlying causal mechanism, could be made a bit more clearly in the manuscript, for instance in the introduction. Of course, more careful formulations such as “evidence for a causal effect” could also be used (as for instance is **almost** done on p. 11:*

“[...] provides evidence for the hypothesis that the impact of social media on individuals varies [...].”

We thank the reviewer for highlighting the issue of casual language; we have followed the discussion with great interest over the last year and can see the immediate benefits for being up-front and transparent about causal claims in the study. Interestingly, Reviewer 1 was of the opposite opinion and wanted us to instead remove all ostensibly causal claims from the manuscript. To find a middle ground, and to ensure our manuscript is not misrepresented in the popular media and press, we have as best possible removed causal claims throughout the manuscript, addressing some of the phrases and terms mentioned by the Reviewer in their comment.

Further we have added a paragraph about the assumptions that would need to be met for our claims to become causal:

“The study has multiple limitations that need to be considered. First, to draw estimates of causal effects from these analyses one would need to adopt the following assumptions: a) there are no time-varying covariates that impact the relation between social media use and life satisfaction; b) the time interval between studies (one year) is the right length to capture the effects of interest; and c) the bidirectional links estimated by our longitudinal model are linear in nature. Only if these assumptions are met can this observational study be said to capture the causal effects between social media and life satisfaction.” (Lines 297-303)

Reviewer 2, Point 2: *Second, with the longitudinal analyses, it is very important to point out that lagged relations (both autoregressive and cross-regressive) are a function of the time-interval between the observations. Hence, if one does not find a cross-lagged relation from X to Y, this should not be taken to mean that X does not have an effect on Y; rather, it implies that it does not have an effect over this time-interval. This point has been raised by many over the years (e.g., by Gollob and Reichardt, 1987; Child Development), and more recently has been discussed by various researchers who advocate a continuous time modeling approach. For instance, Dorman and Griffin (2015; Psychological Methods) very clearly describe this issue in the context of cross-lagged models, and argue that many of our cross-lagged panel studies are based on time intervals that are too long to properly capture the dynamics of the actual underlying process. Specifically, they state: “Our call for researchers to use shorter time lags implies that the common lag of 1 year should be supplemented with shortitudinal studies of much shorter time lags.” When considering social media use and life satisfaction, one could raise the question whether an interval of 1 year is not too long to see any within-person dynamics. Of course, this is a restriction the authors have to work with (as the data were gathered in this way), but some discussion of the (likely) possibility that other results would be found if the interval was 1 month instead of a year, or just 1 week, 1 day, or 1 hour, is important to paint the bigger picture of this kind of research.*

This is a very important point, and we have now added a sentence in the manuscript to acknowledge that the results using shorter time frames could be very different.

“As the outcomes of these longitudinal models depend largely on the time-interval between observations^{41,42}, the annual nature of the data needs to be considered. It is likely that studies on different time frames would show different results and/or reflect distinct mechanisms and processes.” (Lines 202-205)

We also add this as a limitation later in the manuscript:

“The study has multiple limitations that need to be considered. (...) the time interval between studies

(one year) is the right length to capture the effects of interest.” (Lines 297-301)

Reviewer 2, Point 3: *I found Figure 1 rather confusing at first. I think this is because in the description it says “Cross-sectional correlation between estimated social media use and a one-item satisfaction with life measure”; this made me expect to see correlations plotted against something else (i.e., correlation on the y-axis and something else on the x-axis). Instead, what is plotted is the average life satisfaction score for different degrees of social media use at different ages. I think that this could be stated more clearly.*

We agree with the reviewer and have changed the first sentence of the Figure 1 caption to:

“The cross-sectional relationship between social media use and a one-item life satisfaction measure for 72,281 UK participants between the age of 10 and 80 years.” (Lines 119-120)

Reviewer 2, Point 4: *From the description of Figure 3 (which I btw think is a very smart and appealing way to represent these results!), it is not clear whether these “cross-lagged path linking estimated social media use to life satisfaction” is from media use to satisfaction or from satisfaction to social media use; probably the former, but it could be stated more clearly (e.g., saying it is the cross-lagged path from estimated social media use towards life satisfaction one year later).*

Thank you again to the reviewer for this important suggestion. We have changed the first sentence in the Figure 3 caption to:

“Results from the cross-lagged path connecting estimated social media use to life satisfaction one year later, estimated through a Random Intercept Cross-Lagged Panel Model of 17,409 participants (52,556 measurement occasions) aged 10-21.” (Lines 276-278)

Reviewer 2, Point 5: *I am wondering whether how (or if) the inverted U-shaped relations discussed in the context of Figure 1 relate to the results in the longitudinal analyses.*

This is an important point and one which is important to take into account, especially with respect to causal inferences. We have therefore added this as an assumption for causality into our new limitations paragraph:

“First, to draw estimates of causal effects from these analyses one would need to adopt the following assumptions: (...) the bidirectional links estimated by our longitudinal model are linear in nature.” (Lines 297-302)

Reviewer 2, Point 6: *When discussing the comparison between the different life satisfaction measurements at different ages (on p. 19), the authors indicate there was a small inconsistency in the data collection, providing them with 37 cases on which they can test whether there are differences. While I appreciate this point, I think it is also important to indicate that the power to detect differences is not that large with 37 cases (I assume).*

This is a good point, and we ran two power analyses at a significance level of 0.05, and now add the results of these to the text:

“A Welch two-sample t-test comparing the scores of 15 year-olds on both younger adolescent and older adolescent/adult surveys showed no significance difference (Younger Adolescent M = 5.64, N

= 4095; Older Adolescent/Adult M = 5.25, N = 10; $t(3) = 0.33$, $p = 0.77$, power to detect only a large difference of $d = 0.89$). A Bayesian two-sample t-test found anecdotal evidence in favour of the null hypothesis (i.e. no difference) between the two groups ($BF_{10} = 0.49$). A second Welch two-sample t-test further found that 16 year olds showed no significant difference in both surveys (Younger Adolescent M = 5.86, N = 37; Older Adolescent/Adult M = 5.50, N = 4703; $t(37) = 1.74$, $p = 0.09$, power to detect only a medium difference of $d = 0.46$), with a Bayesian two-sample t-test finding anecdotal evidence in favour of the null ($BF_{10} = 0.51$).” (Lines 502-512)

We also now supplement this section with additional analyses that are substantially more highly powered.

Reviewer 2, Point 7: *On p. 19 it is also indicated that additional variables were included in the analyses (more extensive measures of life satisfaction, self esteem, SDQ). It is not clear to me what was done with these variables exactly; were they also analyzed using a RI-CLPM, or where they included in the main RI-CLPM analysis?*

We have now clarified this in the text. We only used these questionnaires for the cross-sectional plot in Figure 2, for Extended Data Figure 4 and analyses of sex differences requested by other reviewers and did not include them in our longitudinal analyses as they were not asked across all ages.

“We also created cross-sectional plots for additional well-being and mental health questionnaires completed only by the younger adolescent sample (age 10-15 years), these measures can be found in Extended Methods 1.” (Lines 525-527)

Reviewer 2, Point 8: *On p. 21 it is stated that there were two variables included as exogenous control variables (time-invariant). Looking at the lavaan code I see these were included as direct predictors of the observed variables, and that their effect was allowed to vary over time (i.e., regression coefficients were not invariant over time). This could be explained in the text. Note that Mulder and Hamaker recently published a manuscript with extensions of the RI-CLPM, which includes a discussion of how to include such time-invariant covariates, and what the effect of different ways of including them are (see <https://www.tandfonline.com/doi/full/10.1080/10705511.2020.1784738>).*

We are grateful for the reviewer for pointing out this unclear area of text. To make it clearer that the control variable is invariant but its effects are estimated freely for each age, we have now changed the text:

“The control variables included in this model are time-invariant mean household income and Index for Multiple Deprivation with freely estimated effects at different ages, to account for the socioeconomic status of both the family and their immediate environment.” (Lines 217-220)

Reviewer 2, Point 9: *On p. 27 the authors discuss certain constraints over time that they tried out; they refer to covariances and variances after age 10; note that in the model you estimate the residual variances and covariances between residuals, when variables are endogenous (dependent); constraining residual variances and covariance does not guarantee that the variances of the (within-person) variables will be invariant over time (this depends on the lagged parameters, and also on the initial variances and covariance).*

We thank the reviewer for pointing out this mistake, and we have now changed the relevant text to:

“constraining the covariance of the residuals of latent factors after age 10 and constraining the

residual variance of both social media and life satisfaction after age 10 ($\chi^2(63) = 2023, p < 0.001$)” (Lines 657-659)

Reviewer 2, Point 10: *On p. 28, the chi-square difference test is discussed for constraining all the cross-lagged paths from life satisfaction to social media use. This test has 21 df, which would mean that it has 21 parameters less than the model without these constraints. Is that correct? I am confused where all these df come from*

We arrive at 21 degrees of freedom because in the fully freed version the model estimates 11 parameters (for each year between 10-21) across both sexes, meaning that it is estimating $11 \times 2 = 22$ parameters. When the model is constrained across both age and sex, it is only estimating 1 parameter, meaning that the difference in parameters is 21.

Reviewer 2, Point 11: *The authors use the term carry-over to refer to the autoregression. Although this is a useful term, I think that it would also be helpful for some readers to indicate this is the same as the (within-person) autoregression.*

We have now included this in the methods section:

“constraining the within-person carry-over effect for both social media estimates and life satisfaction (also known as the within-person autoregression: a positive carry-over means that a person who scores higher than their expected score is more likely to also score higher than their expected score in the following year; $\chi^2(42) = 218, p < 0.001$).” (Lines 660-664)

Reviewer 3

This is a comprehensive and fascinating look at the data on social media use and life satisfaction across age groups. In my view, the paper would be strengthened by some minor revisions to the abstract and text.

Reviewer 3, Point 1: *p. 4: The paper states: “Research has already highlighted gender differences in the links between social media use and well-being in adolescence, although these links have repeatedly been labelled as negligible.”*

This is a strange statement, as several studies examining links between social media use and well-being have found substantial differences, especially among girls (e.g., twice as many heavy users depressed vs. non-users). These studies include:

Kelly, Y., Zilanawala, A., Booker, C., & Sacker, A. (2018). Social media use and adolescent mental health: Findings from the UK Millennium Cohort Study. EClinicalMedicine, 6, 59-68.

Twenge, J. M., & Martin, G. N. (2020). Gender differences in associations between digital media use and psychological well-being: Evidence from three large datasets. Journal of Adolescence, 79, 91-102.

Viner, R. M., Aswathikatty-Gireesh, A., Stiglic, N., Hudson, L. D., Goddings, A-L., Ward, J. L., & Nicholls, D. E. (2019). Roles of cyberbullying, sleep, and physical activity in mediating the effects of social media use on mental health and wellbeing among young people in England: a secondary analysis of longitudinal data. Lancet Child and

Adolescent Health.

These papers should likely be cited, and the phrase about the links being labeled as negligible should be deleted

We thank the reviewer for providing these additional citations and have included them in our introduction. We now recognise that our writing was unclear, as the word “negligible” was not meant to describe the sex differences themselves. We have now clarified this:

“Research has highlighted differences between males and females in the links between social media use and well-being in adolescence in a small subset of the analysed data²⁸, or other datasets^{10,29–33}.”
(Lines 91-93)

Reviewer 3, Point 2: *p 5. How did the authors decide to label 12-15 years as early adolescence and 16-21 years as later adolescence? No citation is given. Or was this division done post-hoc after the results were known? This division is also confusing as the introduction defines adolescence as beginning at age 10, yet here it begins at age 12; the introduction also defines adolescence as lasting until age 24, yet here it ends at age 21.*

We thank the reviewer for raising this inconsistency and pointing us to areas where our writing was unclear. We have now ensured that 10-15 year-olds are consistently referred to as “younger adolescents” in the manuscript. Further, we added an explanation of the age group definitions used into the methods:

“All household members between 10 and 15 years (whom we label here as ‘younger adolescents’) filled out a younger adolescent survey, while those 16 and over filled out an adult survey. 16-21 year-olds (whom we label here as ‘older adolescents’) further completed a short supplement with additional questions.” (Lines: 466-469)

We highlight here that the age groupings were not set up in a post-hoc manner. The specific age boundaries were determined by methodological decisions taken by those coordinating the Understanding Society study. Survey questions needed to be adapted in language and topic to fit those respondents of different ages, and therefore the study organisers designed three different surveys targeted at different age groups. At age 10-15, participants took part in the “youth survey”, at age 16-21 they took part in a “young adult supplement” as well as the “adult survey”, and at age 22+ they only took part in the “adult survey”.

The survey items and the frequency in which they were asked of participants were different across these surveys. For example, annual social media questions were asked in the youth survey and young adult supplement, while adults were only asked about social media use every three years (making it impossible to include those over 21 years in longitudinal modelling). Further extensive mental health questions, e.g. the Strengths and Difficulties Questionnaire, were only given to 10-15 year-olds, meaning that we cannot include those over 15 years in Figure 3 of our study. The age groupings used were therefore informed by *a priori* methodological classifications set up by the Understanding Society study team.

Reviewer 3, Point 3: *p. 6: “This refutes claims that social media displays solely negative relations with well-being ...” No citation is given for this statement. Who has made these claims?*

We thank the reviewer for pointing out that this sentence could be better worded and have now changed it to:

“There was a pronounced inverted U-shaped curve in older adolescence, indicating that those who estimated they engaged in very low or very high social media use reported lower life satisfaction than those who estimated that they used between ‘less than an hour’ and ‘1-3 hours’ of social media a day (i.e., response options ‘2’ and ‘3’). This pattern in between-person associations, where those participants who use the least or the most social media also score lower in well-being, has been previously termed the ‘Goldilocks hypothesis’³⁵.” (Lines 137-143)

Reviewer 3, Point 4: *p. 7 “Younger adolescents showed a different relation than older adolescents: it was less curvilinear and also showed more prominent gender differences in its form.” Describing this association as “less curvilinear” is odd – especially for girls, the association shown in Figure 1 is almost completely linear at ages 12, 13, and 14. Given that, it should probably be described for what it is: linear, not “less curvilinear.”*

To address this point we have completed some model comparisons to test the linearity of the relation between social media and life satisfaction at each age and sex combination. We compared the models using AIC weights (as done in the manuscript for sex differences), which provide us with a preference rating for the completely linear model over the model with the additional squared term (100% means the linear model is totally preferred over the model with the squared term; 50% means the linear model and squared model are equally preferable).

For adolescent girls, the % preference for the linear model is the following: age 10 = 73%, age 11 = 47%, age 12 = 26%, age 13 = 11%, age 14 = 0%, age 15 = 0%, age 16 = 0%, age 17 = 16%, age 18 = 23%, age 19 = 41%, age 20 = 3%, age 14 = 0%.

For adolescent boys, the % preference for the linear model is the following: age 10 = 72%, age 11 = 0%, age 12 = 14%, age 13 = 70%, age 14 = 66%, age 15 = 66%, age 16 = 41%, age 17 = 4%, age 18 = 1%, age 19 = 61%, age 20 = 69%, age 14 = 13%.

As the evidence for a linear model is therefore variable and not exceedingly strong, even at certain ages, we cannot say that the relation is linear per se, however we have changed the phrasing to “more linear” (line 146) as we agree with the reviewer that this would be clearer.

Reviewer 3, Point 5: *In Figure 2A, what is “factor score” on the y-axis? Is this a Z-score or standardized score, or something else? If it’s a Z-score, how were the data standardized? (within study? Within gender?)*

We thank the reviewer for their question and hope our explanation below clarifies any uncertainties. In the Extended Methods section we detail how we estimate a latent factor for each questionnaire and then extract latent factor scores which are an estimate of the true latent factor scores. These latent factor scores are shown in Extended Data Figure 4, Panel A: “In the extension of the cross-sectional analyses, we analysed a range of questionnaires that were only completed by 10-15 year olds in the Understanding Society survey and questionnaires completed by 13 and 14 year olds in the Millennium Cohort Study. As these questionnaires had more than one item, we first extracted latent factor score estimates and applied model comparison to examine measurement invariance across sex before plotting the relation of the latent factor with estimated social media use. (...) After extracting the latent factors for both Understanding Society and the Millennium Cohort Study, we plotted how

they relate to estimated social media use by age (Extended Data Figure 4, Panel A).”

We recognise the axis label was unclear, and have therefore changed it to “Latent Factor Score Estimates”.

Reviewer 3, Point 6: *Another recent paper used the same dataset and came to similar conclusions, yet that paper is not cited here. The authors should say explicitly how their paper is different from the previous research: Booker, C. L., Kelly, Y. J., & Sacker, A. (2018). Gender differences in the associations between age trends of social media interaction and well-being among 10-15 year olds in the UK. BMC Public Health, 18(1), 321.*

We thank the reviewer for raising this. We now include the paper in our introduction, highlighting that our work examines a larger age range 10-21 not just 10-15, something which allows this manuscript to make a central contribution by accounting for age and adolescent development.

“By extending past work to account for age, as a proxy for development, and to encompass the whole adolescent range, this study tests for hypothesised developmental windows of social media sensitivity when stronger links between social media and well-being emerge at specific ages.” (Lines 79-82)

“Research has highlighted differences between males and females in the links between social media use and well-being in adolescence in a small subset of the analysed data²⁸, or other datasets^{10,29–33}. This study therefore also examines potential sex differences in how social media use relates to well-being across adolescent development.” (Lines 91-94)

Reviewer 3, Point 7: *It is unclear why the authors cite reference 5 (Ophir) as an example of “widespread concern” about social media, as this paper was not about concern around social media. Instead, it was focused on a critique of another article – a critique the authors of the original article refuted in their reply. Thus, it seems like an odd choice for a citation.*

This was an error and we have now removed the Ophir citation.

Reviewer 3, Point 8: *The abstract notes that “Decreases in life satisfaction predicted small subsequent increases in social media use.” That suggests life satisfaction changes before social media use. However, the explanation below Figure 3 notes that for certain ages, the link goes the other way: “an individual’s increase from expected levels of estimated social media use predicted a decrease from their expected levels of life satisfaction one year later” – meaning that social media use changes before life satisfaction. The text echoes the figure (p. 9, lines 8-10). The abstract implies that the association is unidirectional (with life satisfaction coming before social media use), but the text and figure show that it is bidirectional, especially at certain ages. The abstract should be brought in line to agree with the text and figure.*

Thank you for raising this unclarity. We have changed the wording slightly to make this clearer:

“Longitudinal analyses of 17,409 participants (10-21 years) highlighted distinct developmental windows of sensitivity to social media in adolescence, when higher estimated social media use predicted a small decrease in life satisfaction one year later (and vice-versa: lower use predicted small increases in satisfaction). These windows occurred at different ages for males (14-15 and 19 years) and females (11-13 and 19 years), possibly suggesting the influence of diverse biological,

psychological and social processes. Decreases in life satisfaction also predicted small subsequent increases in social media use.” (Lines 34-41)

Reviewer 4

This study investigates associations between social media use and life satisfaction in a large sample in the UK. A focus is on age and gender differences, and directional associations are tested in the youth sample. More social media use is associated with lower life satisfaction (although some quadratic effects are observed). Gender effects are found in adolescents but not adults. Bidirectional associations are found in youth, with some evidence that social media use predicts later life satisfaction during specific age periods in adolescent males and females. I appreciate the aims of the analyses, the sample is large, and the authors have done a really nice job with story-telling. However, in my view the authors have not done enough here to test their aims or go beyond what has already been shown in the literature. A number of statements are not supported by statistical tests. Effect sizes are not presented. The directional nature of associations are most interesting, although it is difficult to know what to conclude here – analytic techniques such as causal mediation could be employed to better address questions. Other specific comments below.

We thank the reviewer for their helpful suggestions, which we have addressed in detail below. For example, we have extended the scope of our statistical tests, and we now present the effect sizes clearly in a table (as they were previously only found in the text). Further, we have clarified the assumptions necessary for the bidirectional paths mapped in this study to be interpreted as causal. The study goes beyond the literature in many ways, both in size and age range studied. Further, by using new statistical approaches, we are one of the first to examine age-specific links between social media use and life satisfaction in a longitudinal sample. The resulting findings provide the foundation for new theoretically-informed research approaches (e.g. studies that examine pubertal development and social media use) to understand an internationally pressing research question.

Reviewer 4, Point 1: *The authors use the term gender when I believe they mean “sex” (i.e., when referring to bodily changes in male and female adolescents). The authors state that participants were asked about “sex”. Then why use “gender” throughout?*

We have now changed the variable name to sex as requested, and have included further description of this variable in the methods section:

“Self-reported sex was reported annually (“male”, “female”). When first surveyed, adults and older adolescents were asked to report their sex (options: “male” or “female”); subsequently they were asked to confirm their sex collected in the previous waves. Younger adolescents were asked “are you male or female” (options: “male” or “female”), and allowed to refuse response. If participants’ report of sex varied between waves they were recorded as NA by the survey administrators as part of a cumulative sex variable. Due to the nature of this item design we therefore report the responses as “sex” in this manuscript, however respondents may well have responded according to gender identity or gender assigned at birth based on genitalia as the nature of the questions was ambiguous, especially for younger adolescents. In the waves of data analysed there was however no opportunity to examine to what extent self-report sex was related to gender identity.” (Lines 546-556)

Reviewer 4, Point 2: *Figure 1: Nonlinear associations between social media use and life satisfaction appear to be present in a number of age groups (not just adolescents). Is this the case?*

The reviewer is correct that there are non-linear relations between social media use and satisfaction with life beyond adolescence. However, due to the a priori focus of the manuscript on adolescence, we do not interpret these relations.

Reviewer 4, Point 3: *Figure 1: Do associations hold after controlling for socioeconomic status?*

To address this comment we made a variety of changes that allow us to take into account necessary controls such as SES (measured using household income and neighbourhood deprivation) in our two main cross-sectional analyses.

Firstly, we now replicate our cross-sectional findings displayed in the raw data plot (Figure 1) in analyses that allow us to add the control variables of household income, neighbourhood deprivation (Index of Multiple Deprivation) and year of data collection. Specifically, we run a regression for each age group where social media use and the control variables predict life satisfaction. We then plot the regression coefficient linking social media use to life satisfaction by age in Extended Data Figure 2 and reference it in our main text:

“Furthermore the finding was robust when adding control variables of income, neighbourhood deprivation (measured using the Index of Multiple deprivation) and year of data collection: adolescence was the only time the cross-sectional regression coefficient of social media use predicting life satisfaction fell below -0.1 units (Extended Data Figure 2).” (Lines 111-115)

Extended Data Figure 2: Cross-sectional relation between social media use and life satisfaction across age

The graph plots the regression coefficient predicting life satisfaction from social media use (y axis) for different age groups (x axis). The confidence intervals represent the standard error of the regression coefficient. The grey rectangle highlights the area of the graph where the regression coefficient goes below ± 0.1 units. The regression includes the covariates of household income, neighbourhood deprivation and year of data collection. It shows that the link between social media use and life satisfaction becomes most negative in early adolescence for females rather than males.

Second, we now add the control variables (household income, neighbourhood deprivation and year of data collection) to our analyses of sex differences in the cross-sectional data. Specifically we add them to our regressions using the AIC weights:

“This difference between males and females is statistically supported by an Akaike weights procedure³⁶ showing that models associating estimated social media use and life satisfaction while differentiating for self-reported sex (and controlling for household income, neighbourhood deprivation and year of data collection) were preferred between the ages of 12 and 15 (AIC weights ratios ranging from 5:1 to over 6,000,000:1 in favour of the model differentiating males and females). In contrast, models differentiating for sex were not preferred at other ages (Figure 1, bottom; Extended Data Figure 3).” (151-157)

These analyses including the control variables are now displayed at the bottom of Figure 1 and in the Extended Data.

Reviewer 4, Point 4: *The life satisfaction question for youth vs adults was different, with the youth question asking about happiness (rather than satisfaction). The authors try to address this by comparing small samples of 15 and 16 year olds who received the wrong survey. 15 and 16 year olds do not differ that much in age, so the meaningfulness of this is highly questionable. Therefore, different measures could be a reason for different patterns across the lifespan and should be acknowledged*

We appreciate the concern raised that the life satisfaction question given to 10-15 year olds was different to the question given to those 16 and over. Different questions and surveys were given to different ages because the study spanned a very wide age range from 10 to 80 years. When giving questionnaires to children, problems with language ambiguity or non-comprehension are magnified; every question given needs to be adapted to suit the “cognitive, linguistic and social competence” of the age group (De Leeuw et al., 2004). If this additional care is taken, self-report questionnaires can be successfully administered to those above the age of 7, while adult questionnaires can begin to be administered in older adolescence (e.g. around 18 years, De Leeuw et al., 2004).

With this in mind, the youth survey was designed by Understanding Society (at that point called the British Household Panel Survey) in 1994 to allow for an accurate quantification of well-being of 10-15 year olds. The life satisfaction questionnaire chosen uses a smiley face scale and avoids advanced vocabulary such as “dissatisfied” or “aspects”. The simplification of language has been recommended widely by survey specialists, as children have difficulties responding accurately to questions including vague or advanced words (Holaday & Turner-Henson, 1989). The visual analogue scale is widely

used to further aid children's response to self-report questions, and has been adopted in the well-being and mental health space by the Child Outcomes Research Consortium and the British Association for Counselling and Psychotherapy. Further the youth survey is much shorter than the adult survey because it only asks those questionnaires relevant to early adolescents (i.e. not covering topics such as work, commuting or extensive medical information). The survey in Understanding Society was extensively pre-tested to ensure it was accessible to those aged 10-15 and has been shown to be of high quality by the Understanding Society study team in a quality report (Buck et al., 2006).

In contrast, the wording and response scale of the adult life satisfaction question is harmonised with data collection efforts around the world (e.g. SOEP in Germany) and is written in more advanced language, which those 16 years and over can understand. It omits the smiley face response scale, as this would not be appropriate or necessary for adult participants. It is also part of a longer survey which covers topics only applicable to adults. The differences in question wording, response scales and overall length of survey are therefore important to allow valid responses to be obtained from a large age range.

We have now supplemented our original analysis comparing 15- and 16-year-olds with another quantitative approach to address the Reviewer's concerns. In particular, we have completed an additional analyses that takes into account a larger age range: we examined 13-18 year-olds, i.e. data two years either side of the 15/16 year olds cut-off. To test whether the change in survey significantly changed the life satisfaction scores in this age range we fit a linear regression to predict life satisfaction from both age and a binary dummy variable representing whether the individual took either the youth or adult questionnaire. This showed that, while age significantly predicted life satisfaction scores (which decrease with age, $\beta = -0.09$, $SE = 0.01$, $t \text{ value} = -8.45$, $p < 0.001$), the survey type was not a significant predictor of life satisfaction scores ($\beta = 0.04$, $SE = 0.04$, $t \text{ value} = 1.23$, $p = 0.22$). Further, with 24,513 participants included, this test would be highly sensitive (99% power) to extremely small effects ($f^2 = 0.001$; an f^2 value of 0.02 is small according to Cohen's guidelines) at a conventional alpha level of 0.05. A Bayesian approach comparing a regression including age and the survey category as predictors of life satisfaction scores against a regression only including age as a predictor, preferred the latter with a Bayes Factor of 15.9 (+/- 1.4%). This provides additional strong evidence that the survey category does not impact life satisfaction scores.

There are therefore reasons of both theoretical and psychometric nature to provide children with different life satisfaction questions than adults, and different surveys to complete. Further there is statistical evidence that the responses are comparable. We therefore think that the value of the data far eclipses the differences in questions used: there is no other dataset that we know of which gives us access to life satisfaction and social media data along the entire adolescent age range (10-21 years), making this data extremely valuable to analyse from a research and policy standpoint.

We now detail this in our methods section:

“An irregularity in Understanding Society fieldwork provided us with an opportunity to test that the life satisfaction measurements for the younger adolescent and older adolescent/adult survey were not qualitatively different. Due to a lag in when questionnaires were issued into the field and completed by the participants, 37 16 year-olds mistakenly took the younger adolescent survey while 10 15 year-

olds mistakenly took the older adolescent/adult survey. A Welch two-sample t-test comparing the scores of 15 year-olds on both younger adolescent and older adolescent/adult surveys showed no significance difference (Younger Adolescent $M = 5.64$, $N = 4095$; Older Adolescent/Adult $M = 5.25$, $N = 10$; $t(3) = 0.33$, $p = 0.77$, power to detect only a large difference of $d = 0.89$). A Bayesian two-sample t-test found anecdotal evidence in favour of the null hypothesis (i.e. no difference) between the two groups ($BF_{10} = 0.49$). A second Welch two-sample t-test further found that 16 year olds showed no significant difference in both surveys (Younger Adolescent $M = 5.86$, $N = 37$; Older Adolescent/Adult $M = 5.50$, $N = 4703$; $t(37) = 1.74$, $p = 0.09$, power to detect only a medium difference of $d = 0.46$), with a Bayesian two-sample t-test finding anecdotal evidence in favour of the null ($BF_{10} = 0.51$). Further, we modelled the data for ages 13-18 to predict life satisfaction ratings from both age and a categorical variable indicating the survey type (i.e. younger adolescent or older adolescent/adult survey measurement). While age significantly predicted life satisfaction scores, the type of survey measurement did not (age: $b = -0.09$, $se = 0.01$, $p < 0.001$; survey type: $b = 0.04$, $se = 0.04$, $p = 0.239$; adjusted R-squared = 0.016). With 24,536 participants included, this test would be highly sensitive (99% power) to extremely small effects ($f^2 = 0.001$; an f^2 value of 0.02 is small according to Cohen's guidelines) at a conventional alpha level of 0.05. A Bayesian approach comparing a regression including age and the survey category as predictors of life satisfaction scores against a regression only including age as a predictor, preferred the latter model with a Bayes Factor of 17.0 (+/- 2.1%). These analyses therefore support the conclusion that the measures are not qualitatively different." (Lines 498-523)

Reviewer 4, Point 5: *Where are the statistical tests performed to support claims of sex/gender differences (Figure 2)?*

We have now run model comparisons to examine the differences described in Figure 2, which is now also presented in an extended version in Extended Data Figure 4, using the AIC weights procedure also applied in the manuscript to interpret Figure 1. These analyses now quantitatively support the claims made in the text. We have also added detail in the text describing our findings.

"We supplemented these cross-sectional analyses by examining the longer life satisfaction questionnaire given to 10-15 year-olds in Understanding Society, and 13/14 year-olds in the Millennium Cohort Study (Figure 2): this questionnaire asked about satisfaction with appearance, friends, family, school, schoolwork and life. We found no evidence that a specific sub-component of life satisfaction was the lone driver of the sex differences found in Figure 1. In Understanding Society, sex differences were found predominately for satisfaction with life and appearance (AIC weights: satisfaction with life 71.9%, appearance 69.1%, family 53.8%, friends 59.1%, school 51%, schoolwork 35%). In the Millennium Cohort Study the models differentiating for sex were very highly preferred for all measures (AIC weights: satisfaction with life 100%, appearance 100%, family 100%, friends 99.7%, school 100%, schoolwork 91.2%). Further analyses using a broader range of mental health questionnaires available for these sample participants can be found in Extended Results 1 and Extended Data Figures 4, 5 and 6. The limited age range did not allow us to compare these adolescents with other age groups." (Lines 177-190)

We also add further detail into the Extended Data:

"In Understanding Society data, self-reported well-being showed the largest mean sex differences

across ages 10-15 when compared to questionnaires such as self-esteem or depressive symptoms (mean preference for model differentiating for sex, AIC weights: well-being 72.9%, self-esteem 55.3%, internalising symptoms 57.9%, externalising symptoms 56.2%). In the Millennium Cohort Study, the models differentiating between sex for well-being and self-esteem were highly preferred (100%), while those for externalising symptoms (21.3%) and internalising symptoms (35.5%) were not preferred. For analyses of the correlations see Extended Data Figure 5.

The well-being questionnaire (a questionnaire whose constituent questions include the satisfaction with life question analysed in the main manuscript) therefore seems to show the most substantial sex differences. Examining the constituent sub-questions that make up the wellbeing questionnaire (satisfaction with appearance, family, friends, school, school work and life), we found no evidence that a specific sub-component of life satisfaction was the lone driver of these sex differences (Extended Data Figure 4, Panel B). See the main manuscript and Extended Data Figure 6 for further analyses.”

Reviewer 4, Point 6: *The authors plot associations between social media use and different domains of life satisfaction and mental health (in Figure 2). Then it appears that the authors only eyeball these plots to support claims about gender differences. This seems a missed opportunity to look more specifically at the association between social media use, life satisfaction and mental health. Was social media use statistically differentially related to different domains of life satisfaction? Can poor mental health potentially explain associations?*

We thank the reviewer for pointing out this opportunity. We included the questionnaires to show that life satisfaction does seem to show the greatest sex differences, however as broader well-being measures are only available for ages 10-15 we cannot include them in our longitudinal analyses. Thus more detailed analyses of such items are outside the scope of this manuscript. We however now reference this as an opportunity for further research in our limitations section.

“One could also carry out more targeted investigations, for example, by examining the mental health measures only completed by select age ranges in the datasets (e.g. ages 10-15, displayed in Extended Data Figure 4) to understand how they interrelate over the longitudinal time frame.” (Lines 307-310)

We have also added analyses where we plot the correlation between social media use and the different mental health and life satisfaction outcomes by age and sex (Extended Figures 5 and 6). The error bars showing standard errors allow us to understand how different measures relate to each other.

We add a description of these analyses to the main paper and the Extended Data, as detailed in the reply to Reviewer 4, Point 5.

Extended Data Figure 5: Plot of the cross-sectional correlation between estimated social media use and different mental health and well-being measures (estimated using latent factors) by age and sex

The plot shows that the cross-sectional correlation between estimated social media use and mental health and well-being measures stays relatively stable across early adolescence, except wellbeing whose correlation becomes more negative in females during the period. Using the confidence intervals which represent standard errors one can interpret which measures' relation to social media use is different statistically. For example, across both sexes and most ages, well-being and conduct symptoms are more negatively related to social media use than other measures.

Extended Data Figure 6: Plot of the cross-sectional correlation between estimated social media use and different life satisfaction measures (raw scores) by age and sex

The plot shows that there are differences in how estimated social media use relates to life satisfaction measures across age and sex. Using the confidence intervals that represent standard errors one can interpret which measures' relationship with social media use is different statistically. For example, in females the relationship becomes more negative for satisfaction with appearance, life, school and school work; while the relationship between estimated social media use and satisfaction with family and friends is less negative than other measures and stays relatively stable over age.

Reviewer 4, Point 7: *For question 2, why did the authors control for mean household income, as opposed to income at each wave?*

The large amount of missing data makes it impossible to yearly household income into our longitudinal models, even if we adopt a multiple imputation approach. We have however added this as a limitation to our text.

“We did not include time-varying control variables, such as income, at every wave or year of data collection because the model could not be fitted with the level of missingness present in the data.”
(Lines 652-654)

Reviewer 4, Point 8: *The authors state that “The gender difference in timing suggested that early increases in sensitivity to social media might be due to maturational processes such as puberty, which occur earlier in girls compared*

to boys”. This is overly speculative given that puberty was not investigated.

We agree that we can only speculate that puberty is involved in our findings. However, we believe this speculation is important to raise, especially as it could inform further testing with pubertal measures. Further, the differences in timing between girls and boys is indicative of such a mechanism taking place.

After receiving these reviews we asked around a dozen different labs whether they have data that combined the sort of longitudinal data analysed here with pubertal data but unfortunately such a dataset is currently unavailable. We therefore now highlight the speculative nature of our argument and clearly reference that more work needs to be done to study the relationship between pubertal development, social media use and wellbeing:

“Speculatively, the sex difference in timing suggested that early increases in sensitivity to social media might be due to maturational processes such as puberty, which occur earlier in girls compared to boys²⁰. A later increase in sensitivity to social media, which was present at age 19 for both young men and young women, suggested a different underlying process may be at play in older adolescence [...] Speculatively, this might be related to changes in the social environment such as a move away from home and subsequent disruptions in social networks. However, these explanations cannot be tested directly in this dataset and require further targeted investigation using data containing pubertal and social measurements.” (Lines 262-272)

Reviewer 4, Point 9: *Were statistical tests performed to support claims of gender differences in Figure 3?*

The statistical tests for Figure 3 were done via model comparison, i.e. comparing a model that constrains the impact of social media on life satisfaction to be equal across age and sex with a model that allows the impact to vary. This was done via a chi-squared test procedure, as well as information criteria comparison, that is detailed in the methods section. The results that support the sex differences are detailed in the results section:

“A model constraining the path of life satisfaction predicting estimated social media use to be constant across age and sex, while allowing the path of estimated social media use predicting life satisfaction to vary, was highly preferred over other model constraints (AIC weights procedure: 99.2%, see Extended Data Figure 8; model fit: $\chi^2(455) = 1,234.74$, $p < 0.001$, RMSEA = 0.014, [0.013, 0.015], CFI = 0.95, SRMR = 0.072).” (Lines 225-231)

Reviewer 4, Point 10: *Were the prospective effects of social media use on life satisfaction weaker/stronger than those of life satisfaction on social media use?*

The standardised cross lagged path of life satisfaction -> social media use was -0.02 - -0.03. The cross lagged path from social media -> life satisfaction dipped below this lower CI of -0.03 only at age 11, 12, 13, 14, 16, 19 and 20 for girls and age 10, 11, 13, 14, 15 and 19 for boys. As the difference between both paths is therefore age-dependent, and only present in very select age/sex groups, it is not possible to make a generalised statement about whether the paths were bigger or smaller.

Reviewer 4, Point 11: *Please state effect sizes for all findings.*

The effect sizes are displayed throughout the manuscript as the standardised “beta” coefficients when reporting the RI-CLPMs. These are standardised coefficients and therefore represent the constituent effect sizes. To further highlight the effect sizes we now provide a table with the standardised coefficients, which should help the reader gain an overview of the results.

Table 1: Results of the best fitting Random-Intercept Cross Lagged Panel Model examining the link between life satisfaction and social media use across ages 10-21

	b (regression coefficient)	Standard Error	β (standardised effect size)	p
Life Satisfaction -> Social Media Use				
Invariant across age and sex	-0.02 [-0.03, -0.01]	0.007	-0.02--0.03	0.004
Social Media Use -> Life Satisfaction (Female)				
Age 10	0.02 [-0.09, 0.12]	0.05	0.01	0.753
Age 11	-0.11 [-0.21, -0.02]	0.05	-0.09	0.020
Age 12	-0.14 [-0.22, -0.07]	0.04	-0.12	0.000
Age 13	-0.08 [-0.15, -0.01]	0.03	-0.07	0.019
Age 14	-0.04 [-0.11, 0.03]	0.03	-0.04	0.215
Age 15	0.01 [-0.07, 0.10]	0.04	0.01	0.784
Age 16	-0.07 [-0.15, 0.01]	0.04	-0.06	0.080
Age 17	0.00 [-0.08, 0.08]	0.04	0.00	0.937
Age 18	0.02 [-0.06, 0.10]	0.04	0.02	0.642
Age 19	-0.16 [-0.25, -0.07]	0.05	-0.13	0.000
Age 20	-0.06 [-0.14, 0.03]	0.04	-0.05	0.181
Social Media Use -> Life Satisfaction (Male)				
Age 10	-0.06 [-0.16, 0.05]	0.05	-0.04	0.291
Age 11	-0.05 [-0.15, 0.04]	0.05	-0.04	0.275
Age 12	0.02[-0.06, 0.09]	0.04	0.01	0.708
Age 13	-0.05[-0.12, 0.03]	0.04	-0.04	0.202
Age 14	-0.10 [-0.17, -0.03]	0.04	-0.10	0.005

Age 15	-0.18 [-0.29, -0.08]	0.05	-0.12	0.001
Age 16	0.02 [-0.05, 0.10]	0.04	0.02	0.551
Age 17	0.01 [-0.07, 0.09]	0.04	0.01	0.806
Age 18	-0.00 [-0.08, 0.08]	0.04	-0.00	0.956
Age 19	-0.16 [-0.26, -0.07]	0.05	-0.13	0.001
Age 20	0.02 [-0.06, 0.11]	0.04	0.02	0.614

Reviewer 4, Point 12: *It is stated on one section that latent factors were measured for the SDQ, but in another section is stated that the emotion and conduct subscale scores were used. Please clarify.*

We have clarified this in Extended Data Methods 1:

“The SDQ is a commonly used and widely validated measure for psychosocial functioning applied in school, home and clinical environments⁵. It encompasses five questions each about emotional symptoms, conduct problems, hyperactivity/inattention, peer relationship problems and prosocial behaviour (0 = not true, 1 = somewhat true, 2 = certainly true). We analysed conduct scores as a measure of externalising symptoms and emotional scores as an internalising symptoms measure.”

“We extracted two latent factors for SDQ emotional/internalising symptoms and conduct/externalising symptoms respectively”

“We extracted estimates for two latent factors from the SDQ: internalising symptoms (emotional subscale) and externalising symptoms (conduct subscale) (see methods above).”

Reviewer 5

This manuscript examined the evidence on the relationship between social media and life satisfaction during adolescence using data from the UKHLS and Millennium Cohort Study. As the authors comment, this is a much disputed field and new evidence is always welcome. It extends previous work on gender differences by adding a longitudinal perspective and a different outcome, life satisfaction. The authors conclude that social media use predicts a small decrease in life satisfaction one year later, occurring at different ages depending on gender, and that decreases in life satisfaction also predicted small subsequent increases in social media use.

Reviewer 5, Point 1: *The abstract is quite confusing as it first mentions an age range of 10-80 years. Then reports that the cross-sectional relationship between estimated social media use and life satisfaction was more negative in adolescence (12-15 years), compared to other age groups. Then longitudinal analyses are based on 10-21 year olds. Why all these different age ranges?*

We understand that in the abstract and throughout the whole manuscript, the age ranges were

unclear. We have therefore added an explanation of the age group definitions used into the methods:

“All household members between 10 and 15 years (whom we label here as ‘younger adolescents’) filled out a younger adolescent survey, while those 16 and over filled out an adult survey. 16-21 year-olds (whom we label here as ‘older adolescents’) further completed a short supplement with additional questions.” (Lines: 466-469)

Different age ranges are present due to the nature of the data; they were not set up in a post-hoc manner. The specific age boundaries were determined by methodological decisions taken by those coordinating the Understanding Society study. Survey questions needed to be adapted in language and topic to fit those respondents of different ages, and therefore the study organisers designed three different surveys targeted at different age groups. At age 10-15, participants took part in the “youth survey”, at age 16-21 they took part in a “young adult supplement” as well as the “adult survey”, and at age 22+ they only took part in the “adult survey”.

The survey items and the frequency in which they were asked of participants were different across these surveys. For example, annual social media questions were asked in the youth survey and younger adult supplement, while adults were only asked about social media use every three years (making it impossible to include those over 21 years in the longitudinal modelling). Further extensive mental health questions, e.g. the Strengths and Difficulties Questionnaire, were only given to 10-15 year-olds, meaning that we cannot include over 15 year-olds in Figure 2 of our study.

The cross-sectional analyses were therefore done on all the data (10-80 years), finding that younger adolescents show the greatest sex difference. This was then followed up by an analysis on the data that could be used for longitudinal analyses (10-21 years). This is not possible to explain in the strict word limits of the abstract, however we have now clarified this in the paper.

Reviewer 5, Point 2: *From figure 1, it looks to me as though estimated social media use and life satisfaction was more negative at age 11 too. I could find no mention on how the conclusion that the correlation was more negative at ages 12-15 years was made, only that there were tests for gender differences at different ages. Secondly, in what way did social media use predicted a decrease in life satisfaction? Was it higher or lower SMU?*

The confidence intervals were very wide at age 11 and we therefore decided not to interpret them. We added an explanation of this into our caption of Figure 1: “It should be noted that as high levels of social media use are very rare in the youngest and oldest age groups present in the data (e.g. ages 10, 11 and 60+, Extended Data Figure 1), we cannot evaluate functional form in these groups”. Further, as the figure only shows correlations that are cross-sectional, we cannot make any claims of the directionality of the relationship. We have added further explanation of this in the Figure caption:

“The cross-sectional relationship between social media use and a one-item life satisfaction measure for 72,287 UK participants between the age of 10 and 80 years.” (Lines 119-120)

Reviewer 5, Point 2: *The differences in the measurement of life satisfaction in the youth and adult surveys raise many questions. I am not convinced by the Welch two-sample t-tests when there are such large differences in sample sizes for the two groups. The items on overall life satisfaction come after other individual items which vary between the youth and adult surveys and must differentially prime the respondents for the overall question*

We appreciate the concern raised that the life satisfaction question given to 10-15 year olds was different to the question given to those 16 and over, and that the surveys in which they were imbedded differed. Different questions and surveys were given to different ages because the study spanned a very wide age range from 10 to 80 years. When giving questionnaires to children, problems with language ambiguity or non-comprehension are magnified; every question given needs to be adapted to suit the “cognitive, linguistic and social competence” of the age group (De Leeuw et al., 2004). If this additional care is taken, self-report questionnaires can be successfully administered to those above the age of 7, while adult questionnaires can begin to be administered in older adolescence (e.g. around 18 years, De Leeuw et al., 2004).

With this in mind, the youth survey was designed by Understanding Society (at that point called the British Household Panel Survey) in 1994 to allow for an accurate quantification of well-being of 10-15 year olds. The life satisfaction questionnaire chosen uses a smiley face scale and avoids advanced vocabulary such as “dissatisfied” or “aspects”. The simplification of language has been recommended widely by survey specialists, as children have difficulties responding accurately to questions including vague or advanced words (Holaday & Turner-Henson, 1989). The visual analogue scale is widely used to further aid children’s response to self-report questions, and has been adopted in the well-being and mental health space by the Child Outcomes Research Consortium and the British Association for Counselling and Psychotherapy. Further the youth survey is much shorter than the adult survey because it only asks those questionnaires relevant to early adolescents (i.e. not covering topics such as work, commuting or extensive medical information). The survey in Understanding Society was extensively pre-tested to ensure it was accessible to those aged 10-15 and has been shown to be of high quality by the Understanding Society study team in a quality report (Buck et al., 2006).

In contrast, the wording and response scale of the adult life satisfaction question is harmonised with data collection efforts around the world (e.g. SOEP in Germany) and is written in more advanced language, which those 16 years and over can understand. It omits the smiley face response scale, as this would not be appropriate or necessary for adult participants. It is also part of a longer survey which covers topics only applicable to adults. The differences in question wording, response scales and overall length of survey are therefore important to allow valid responses to be obtained from a large age range.

We have now supplemented our original analysis comparing 15- and 16-year-olds with another quantitative approach to address the Reviewer’s concerns. In particular, we have completed an additional analyses that takes into account a larger age range: we examined 13-18 year-olds, i.e. data two years either side of the 15/16 year olds cut-off. To test whether the change in survey significantly changed the life satisfaction scores in this age range we fit a linear regression to predict life satisfaction from both age and a binary dummy variable representing whether the individual took either the youth or adult questionnaire. This showed that, while age significantly predicted life satisfaction scores (which decrease with age, $\beta = -0.09$, $SE = 0.01$, $t \text{ value} = -8.45$, $p < 0.001$), the survey type was not a significant predictor of life satisfaction scores ($\beta = 0.04$, $SE = 0.04$, $t \text{ value} = 1.23$, $p = 0.22$). Further, with 24,513 participants included, this test would be highly sensitive (99% power) to extremely small effects ($f^2 = 0.001$; an f^2 value of 0.02 is small according to Cohen’s guidelines) at a conventional alpha level of 0.05. A Bayesian approach comparing a regression including age and the survey category as predictors of life satisfaction scores against a regression only

including age as a predictor, preferred the latter with a Bayes Factor of 15.9 (+/- 1.4%). This provides additional strong evidence that the survey category does not impact life satisfaction scores.

There are therefore reasons of both theoretical and psychometric nature to provide children with different life satisfaction questions than adults, and different surveys to complete. Further there is statistical evidence that the responses are comparable. We therefore think that the value of the data far eclipses the differences in questions used: there is no other dataset that we know of which gives us access to life satisfaction and social media data along the entire adolescent age range (10-21 years), making this data extremely valuable to analyse from a research and policy standpoint.

We now detail this in our methods section:

“An irregularity in Understanding Society fieldwork provided us with an opportunity to test that the life satisfaction measurements for the younger adolescent and older adolescent/adult survey were not qualitatively different. Due to a lag in when questionnaires were issued into the field and completed by the participants, 37 16 year-olds mistakenly took the younger adolescent survey while 10 15 year-olds mistakenly took the older adolescent/adult survey. A Welch two-sample t-test comparing the scores of 15 year-olds on both younger adolescent and older adolescent/adult surveys showed no significance difference (Younger Adolescent M = 5.64, N = 4095; Older Adolescent/Adult M = 5.25, N = 10; $t(3) = 0.33$, $p = 0.77$, power to detect only a large difference of $d = 0.89$). A Bayesian two-sample t-test found anecdotal evidence in favour of the null hypothesis (i.e. no difference) between the two groups (BF10 = 0.49). A second Welch two-sample t-test further found that 16 year olds showed no significant difference in both surveys (Younger Adolescent M = 5.86, N = 37; Older Adolescent/Adult M = 5.50, N = 4703; $t(37) = 1.74$, $p = 0.09$, power to detect only a medium difference of $d = 0.46$), with a Bayesian two-sample t-test finding anecdotal evidence in favour of the null (BF10 = 0.51). Further, we modelled the data for ages 13-18 to predict life satisfaction ratings from both age and a categorical variable indicating the survey type (i.e. younger adolescent or older adolescent/adult survey measurement). While age significantly predicted life satisfaction scores, the type of survey measurement did not (age: $b = -0.09$, $se = 0.01$, $p < 0.001$; survey type: $b = 0.04$, $se = 0.04$, $p = 0.239$; adjusted R-squared = 0.016). With 24,536 participants included, this test would be highly sensitive (99% power) to extremely small effects ($f^2 = 0.001$; an f^2 value of 0.02 is small according to Cohen’s guidelines) at a conventional alpha level of 0.05. A Bayesian approach comparing a regression including age and the survey category as predictors of life satisfaction scores against a regression only including age as a predictor, preferred the latter model with a Bayes Factor of 17.0 (+/- 2.1%). These analyses therefore support the conclusion that the measures are not qualitatively different.” (Lines 498-523)

Reviewer 5, Point 3: *Why were 12-15 year olds and 16-21 year olds compared? Justification would be helpful since the UN define adolescence as the period between 10 and 19 years of age. The WHO also defines adolescents as individuals aged 10-19 years. So why drop the 10-11 year olds in UKHLS and extend adolescence to 21? A more hypothesis driven analysis would have usefully dispelled thoughts of cherry-picking results*

We now understand that the age groupings used in our manuscript were unclear and at times inconsistent. We have therefore ensured that 10-15 year-olds are now consistently referred to as “younger adolescents” in the manuscript, and that these are compared as a group. Further, we added an explanation of the age group definitions used into the methods:

“All household members between 10 and 15 years (whom we label here as ‘younger adolescents’) filled out a younger adolescent survey, while those 16 and over filled out an adult survey. 16-21 year-olds (whom we label here as ‘older adolescents’) further completed a short supplement with additional questions.” (Lines: 466-469)

For more information, see our reply to Reviewer 5, Point 1.

Reviewer 5, Point 4: *Figure 1 suggests that the age-period-cohort problem could be present in the data. For example, mean SMU/LS correlation at an 18 year old girl's 1st observation is ~5, but mean SMU/LS correlation at a 13 year old girl's last observation is more like 4.5. Yet there is no adjustment for birth year or wave in the longitudinal analyses*

We have now added in cross-sectional analyses that include year of data collection and age in the model, addressing this issue. Extended Data Figure 2 shows the cross-sectional relation between social media use and life satisfaction by age, while controlling for household income, neighbourhood deprivation and year of data collection.

Extended Data Figure 2: Cross-sectional relation between social media use and life satisfaction across age

The graph plots the regression coefficient predicting life satisfaction from social media use (y axis) for different age groups (x axis). The confidence intervals represent the standard error of the regression coefficient. The grey rectangle highlights the area of the graph where the regression coefficient goes below ± 0.1 units. The regression includes the covariates of household income, neighbourhood deprivation and year of data collection. It shows that the link between social media use and life satisfaction are most negative in early adolescence for females rather than males.

The large amount of missing data makes it impossible to add year of data collection or year of birth

as a time varying covariate into our longitudinal models, even if we adopt a multiple imputation approach. We have however added this as a limitation to our text.

“We did not include time-varying control variables, such as income, at every wave or year of data collection because the model could not be fitted with the level of missingness present in the data.” (Lines 652-654)

Reviewer 5, Point 5: *If you refer to male/female, then you are really talking about sex not gender. If you are using the term gender to mean that you are interested in the socially constructed influences on social media behaviour and life satisfaction, then it would be best to use the terms men/women throughout irrespective of the labels used in the original surveys. Alternatively, using male/female terminology infers that you believe that biological influences are at play despite standing on the fence with “biological, cognitive or social” (page 4, line 14). Which is it?*

We have now changed the variable name to sex as requested, and have included further description of this variable in the methods section:

“Self-reported sex was reported annually (“male”, “female”). When first surveyed, adults and older adolescents were asked to report their sex (options: “male” or “female”); subsequently they were asked to confirm their sex collected in the previous waves. Younger adolescents were asked “are you male or female” (options: “male” or “female”), and allowed to refuse response. If participants’ report of sex varied between waves they were recorded as NA by the survey administrators as part of a cumulative sex variable. Due to the nature of this item design we therefore report the responses as “sex” in this manuscript, however respondents may well have responded according to gender identity or gender assigned at birth based on genitalia as the nature of the questions was ambiguous, especially for younger adolescents. In the waves of data analysed there was however no opportunity to examine to what extent self-report sex was related to gender identity.” (Lines 546-556)

Reviewer 5, Point 6: *I did not understand the model selection strategy. First, I did not notice any mention of the estimation method. I am less familiar with the *r* SEM implementation than other statistical packages. Is the chi-square test appropriately estimated, depending on the estimation method? This seems important, as the BIC statistics favour simpler measurement models than those selected by the authors based on the chi-sq difference test. Second, the authors move on to use AIC weights for the comparison of the longitudinal models. Would the same conclusions have been drawn if BIC weights were used instead?*

We thank the reviewer for their questions and are happy to provide some more explanation. We noted the details of our estimation methodology in the methods section: “The model was fit using robust Maximum Likelihood (MLR) and robust Huber-White Standard errors, and missing data was accounted for by Full-Information Maximum Likelihood estimation”. We use robust Maximum Likelihood (MLR) with robust Huber-White Standard errors, which has been shown to be optimal for large datasets with non-normal data. This yields unbiased and efficient estimation under the assumption of MAR, and has been shown to outperform other techniques such as least squares (Schermele-Engel et al., 2003). Robust ML requires slightly larger sample sizes than standard ML, with guidelines varying from 400 (Boomsma & Hoogland, 2001) to $N \geq 2,000$ (Yang-Wallentin & Jöreskog, 2001) – our sample is considerably larger than even the most stringent thresholds, confirming that the chi square statistic is appropriately estimated.

The choice between AIC and BIC has a rich literature comparing the relative strengths and weaknesses (Vandekerckhove et al., 2015, Burnham & Anderson, 2004, Kuha, 2004, Vrieze, 2012). The asymptotic properties differ, with the AIC guaranteeing the selection of the model that best explains unseen data (and as such is asymptotically equivalent to cross-validation), whereas the BIC guarantees the selection of the true model, if the true model is among the candidate models. For our purposes, our models and data (e.g. temporal frequency) are, by necessity, a simplified approximation of the (continuous) underlying and unfolding processes, and our goal is to most closely approximate the subtle impacts and how they change over time – for that reason, we favour the AIC (see also Vrieze, 2012, Yang, 2005). Although opinions differ on this point, arguments have been put forth that ‘*There is a clear philosophy, a sound criterion based in information theory, and a rigorous statistical foundation for AIC*’ (Burnham & Anderson, 2004, p. 261). For these reasons, on balance we believe the AIC is best suited to our a priori conceptualization and goals.

- Boomsma, A., & Hoogland, J. J. (2001). The robustness of LISREL modeling revisited. *Structural equation models: Present and future. A Festschrift in honor of Karl Jöreskog*, 2(3), 139-168.
- Burnham, K. P.; Anderson, D. R. (2004), "Multimodel inference: understanding AIC and BIC in Model Selection" (PDF), *Sociological Methods & Research*, 33: 261–304,
- Kuha, J. (2004). AIC and BIC: Comparisons of assumptions and performance. *Sociological methods & research*, 33(2), 188-229.
- Schermelleh-Engel, K., Moosbrugger, H., & Müller, H. (2003). Evaluating the fit of structural equation models: Tests of significance and descriptive goodness-of-fit measures. *Methods of psychological research online*, 8(2), 23-74.
- Vandekerckhove, J., Matzke, D., & Wagenmakers, E. J. (2015). Model comparison and the principle of parsimony (pp. 1-29). *The Oxford handbook of computational and mathematical psychology*, 300.
- Vrieze, S. I. (2012). Model selection and psychological theory: a discussion of the differences between the Akaike information criterion (AIC) and the Bayesian information criterion (BIC). *Psychological methods*, 17(2), 228.
- Yang, Y. (2005), "Can the strengths of AIC and BIC be shared?", *Biometrika*, 92: 937–950, [doi:10.1093/biomet/92.4.937](https://doi.org/10.1093/biomet/92.4.937).
- Yang-Wallentin, F., & Jöreskog, K. G. (2001). Robust standard errors and chi-squares for interaction models. *New developments and techniques in structural equation modeling*, 159-171.

Reviewer 5, Point 7: *There was no discussion of the limitations of the study. I believe this is necessary as this is an observational study and there appears to be some subjectivity in the methods employed (see below)*

We have now added a specific limitations section, where we discuss observational data and causality.

“The study has multiple limitations that need to be considered. First, to draw estimates of causal effects from these analyses one would need to adopt the following assumptions: a) there are no time-varying covariates that impact the relation between social media use and life satisfaction; b) the time interval between studies (one year) is the right length to capture the effects of interest; and c) the bidirectional links estimated by our longitudinal model are linear in nature. Only if these assumptions

are met can this observational study be said to capture the causal effects between social media and life satisfaction. Second, the data are self-report and therefore only allow inferences about the impact of self-estimated time on social media, rather than objectively measured social media use. Finally, this paper is a springboard for further mechanistic work, for example, in datasets with pubertal or additional social measurements. One could also carry out more targeted investigations, for example, by examining the mental health measures only completed by select age ranges in the datasets (e.g. ages 10-15, displayed in Extended Data Figure 4) to understand how they interrelate over the longitudinal time frame.” (Lines 297-310)

Reviewer 5, Point 8: *There is also no discussion of how these new findings fit with or contradict findings from previous work on the relationship between SMU and adolescent wellbeing. Do the longitudinal findings confirm or refute findings from cross-sectional analyses? Do relationships with life satisfaction agree with other dimensions of wellbeing such as depression? Are the supplementary analyses using internalising and externalising symptoms congruent with previous research with these outcomes?*

We thank the reviewer for challenging us to further position our results in the context of the literature. While the age-sensitive effects are something very new, and therefore do not refute or support previous work, we have now added more detail about how our findings contribute to the literature:

“While the results support past longitudinal work finding bidirectional influences between social media use and life satisfaction¹⁰, it also goes beyond that to show that such influences inherently vary due to processes such as development.” (Lines 317-320)

As we do not have the space to discuss the further well-being items in more detail, we have moved these into the supplementary material (Extended Data Figures 4, 5 and 6).

Reviewer 5, Point 9: *More important that the “72,281 10-80 year-old participants” (page 5, line 9), is information on the number of younger and older adolescents, and the average number of observations*

It is difficult to say the number of unique participants in each category as participants age into and out of each category every year, and so we are at the risk of counting participants twice if we reported them separately. However, we have now included the number of adolescents used in the longitudinal analyses:

“We applied this modelling framework to data provided by 10 to 21 year-olds in Understanding Society, who completed estimated social media use and life satisfaction measures annually for up to seven waves (17,409 participants)^{37,38}” (Lines 200-202)

Further, in the methods section we record the number of participants within the waves:

“These exclusions left a sample of 72,287 participants over 7 waves (1 wave = 12,444 participants, 2 waves = 8,777 participants, 3 waves = 8,794 participants, 4 waves = 7,426 participants, 5 waves = 7,099 participants, 6 waves = 9,898 participants, 7 waves = 17,849 participants).” (Lines 599-602)

We also now report the number of observations for each age during adolescence in the supplementary materials:

Age	Number of Measurement Occasions
10	3895
11	4048
12	4149
13	4229
14	4272
15	4105
16	4740
17	4889
18	4804
19	4641
20	4435
21	4349

Extended Data Table 1: Number of measurement occasions by age in younger and older adolescents (ages 10-21)

Reviewer 5, Point 10: *Can you confirm the age range for “late adolescence” (page 6, line 20).*

To be consistent with the categories described in our methods, we have added the age range of 16-21 years to the text:

“The relations in the raw data also differed when comparing younger (10-15 years) and older adolescents (16-21 years; for more information about these a priori age categories see methods). There was a pronounced inverted U-shaped curve in older adolescence...” (Lines 135-137)

Reviewer 5, Point 11: *Discussing the model constrains on page 10, “A model constraining the path of life satisfaction predicting estimated social media use to be constant across age and gender, while allowing the path of estimated social media use predicting life satisfaction to vary, was highly preferred over other model constraints”. Surely more to the point, is whether the constrained model fits as well as an unconstrained model?*

The text references Extended Data Figure 8, which shows four different types of model (completely unconstrained, completely constrained, only constraining the cross-lagged path from social media use to life satisfaction and only constraining the cross-lagged path from life satisfaction to social media use). We find that the last model (only constraining the cross-lagged path) fits better than all the other three, i.e. also the completely unconstrained or constrained model.

Reviewer 5, Point 12: *Could it also be that case that higher LS predicted a negative change in SMU rather than a lower than expected LS score in one year predicted a positive change in SMU (page 10, line 24)? See also the abstract: in what way did social media use predicted a decrease in life satisfaction? Was it higher or lower SMU?*

Yes, it could be either way and the model cannot give evidence for one direction over and above the other. We have now added a “for example” and “vice versa” to the phrase to make this clearer.

“The constrained cross-lagged path of life satisfaction predicting estimated social media use was negative, meaning that, for example, if an individual scored lower than their expected value of life satisfaction in one year this predicted a positive change from their expected estimated social media use one year later (or vice versa, $b = -0.02$ [$-0.03, -0.01$], $se = 0.007$, $\beta = -0.02$ – -0.03 , $p = 0.004$).” (Lines 236-240)

We also added “(and vice-versa: lower use predicted small increases in satisfaction)” (Lines 37-38) to the abstract.

Reviewer 5, Point 13: *Can it be clarified what the age range is for “young adults” (page 20, line 8). It is slightly confusing as the methods text suggests this is 16-21 (page 18, line 10) but figure 1 shows detail for ages 20-25 and 26-29 and 10 year bands thereafter. What is special about the 20s?*

We have now clarified that Understanding Society calls the additional survey for 16-21 year olds the “young adult supplement” but that we label this age group “older adolescents” as this fits with the definition of adolescence we use, following Sawyer et al 2018. We have changed the manuscript to align to this, for example:

“All household members between 10 and 15 years (whom we label here as ‘younger adolescents’) filled out a younger adolescent survey, while those 16 and over filled out an adult survey. 16-21 year-olds (whom we label here as ‘older adolescents’) further completed a short supplement with additional questions.” (Lines: 466-469)

We decided to display the 20s in two five-year bands because of its proximity with adolescence, while the rest of adulthood is presented in 10-year tranches.

Reviewer 5, Point 14: *When age was defined (page 21, line 9), did the authors use the interview date or the sampling wave/year?*

Age was provided directly by the Understanding Society team, and was defined as age at interview date. We have now added more detail about the age variable to the text:

“Age for both the adolescent samples as well as the adult samples were derived by the data provider using self-reported date of birth and the interview date.” (Lines 558-559)

Reviewer 5, Point 15: *Why exclude the 15 and 16 year olds who filled out the incorrect survey when you have previously reported that it did not make any difference whether the youth or adult questionnaire was used?*

We agree with the reviewer and have now included these participants in the analyses, except those of Figure 2 because these can only be done on data from the youth questionnaire.

Reviewer 5, Point 16: *I presume that participants who completed a questionnaire twice in one age category must have had a late interview in one wave and an on-time interview in the next wave. If so, this addresses my previous question about defining age. But if age is calculated using year/month information and then rounded to the nearest year, then 9k respondents would not have to be dropped*

As age was provided by Understanding Society, we asked their team for more information on this. They answered that data collection for each wave takes place over a two-year period, and the annual waves are therefore overlapping. It can therefore be the case that a participant gets surveyed twice in one year. As our statistical models do not work if participants have two responses in only one age group, we need to remove the duplications from our analyses.

Reviewer 5, Point 17: *If sampling weights were not applied (why?) then would it not have been possible to have at least adjusted all models using the sampling strata variables?*

We agree that having sampling weights would be a welcome addition to the manuscript. However, the sampling weights provided by Understanding Society are given per year (i.e. sampling weight for 2018 for example). Due to the innovative and unusual nature of our data analysis (examining data by age rather than by year), the sampling weights do not correspond to our analytical design. We therefore instead note the lack of sampling weights as a limitation in our study:

“Due to the nature of our modelling approach we were not able to integrate sampling weights into our estimation strategy. This limits the extent to which we can generalize our findings to the whole UK population.” (Lines 608-610)

Summary

We thank the reviewers for taking the time to consider our manuscript. We are convinced the work has improved substantially, and we hope you agree that it will make an important contribution to the literature around this topic.

Yours Sincerely,

Amy Orben

College Research Fellow
Emmanuel College & MRC Cognition and Brain Sciences Unit
University of Cambridge

Reviewers' comments:

Reviewer #1 (Remarks to the Author):

The authors have adequately addressed all of my main criticisms in their revisions to the article and in their detailed and thorough response letter to the Reviewers.

Candice Odgers
Professor of Psychological Science
University of California, Irvine

Reviewer #2 (Remarks to the Author):

Review for "Windows of developmental sensitivity to social media" -- revision 1

I appreciate the changes that the authors have made. Clearly, my suggestions (reviewer 2) and the comments made by reviewer 1 in the previous round with respect to the use of causal language, were difficult to reconcile. While I still stand by my earlier suggestions that it would be helpful for psychology and related fields to be more open and upfront about one's interest in causal relations (even or especially in the absence of a randomized controlled trial), I understand the choices the authors have made and will not further push this matter.

I have only a few mostly textual comments that could be easily clarified/solved:

Line 35: "highlighted"; perhaps a better term in combination with "sensitivity" (which suggests a causal relation), is "suggested"

Lines 104-107: You indicate that the participants were measured up to 7 times; does this mean that the same individual is included multiple times in the data that were used to generate Fig 1, or was only one of these occasions used per person?

Lines 142-143: "Goldilocks hypothesis"; I think a brief sentence, stating that this refers to the idea that too much or too little are both suboptimal, would be helpful (also for foreign readers that may not know this story or do not recognize the English name of it?).

Lines 151-157, and elsewhere: You say you use AIC weights, but from the main text it is not really clear what this is and how it was used. I am not suggesting a major explanation should be given, but it would be helpful if some intuition is provided. Personally, I would have thought these are the same as Akaike weights, meaning that a set of models is compared and each model is given a probability of being the right (or best) model, and these probabilities add to 1. But in lines 183-187, these weights are used to present percentages, and these do not add up to 100, so I am not sure what the weights do and how

they should be interpreted then (or how these percentages should be interpreted). Furthermore, I find the word “preferred” on p. 7 somewhat awkward; I wouldn’t say that the AIC (weights) prefer a particular model, but rather that we prefer a model based on the fit measures or weights, or that we select a model based on them; but this may be a matter of taste.

Lines 299-300: “there are no time-varying covariates that impact the relation between social media use and life satisfaction” – I would be tempted to use the term “time-varying unobserved confounders”, to emphasize the causal nature of the effect of such a variable (although “impact” of course is also a causal term). Also, I think another assumption one needs to make here is that there is no measurement error in the variables.

Lines 322-324: “While the windows of sensitivity to social media are prominent in aggregate, they will most probably also meaningfully differ across individuals, as each person’s sensitivity is further influenced by a wide range of individual, peer and environmental dynamics.”; I would also say that a further line of research could be into what kind of social media use one is involved in (Facebook may have a different effect than twitter or TikTok), and how one uses it; for instance, there may be a difference in the impact based on whether one is more actively involved (e.g., posting and then seeing how many likes one gets), or more passively (e.g., mostly just looking/reading and liking posts of others), and participating in WhatsApp groups may be different than using WhatsApp to communicate with individual friends.

Ellen Hamaker

Reviewer #3 (Remarks to the Author):

I thank the authors for their detailed response letter. They have satisfactorily responded to my previous concerns.

Reviewer #4 (Remarks to the Author):

4.2. In the manuscript, authors state: “To investigate the existence of developmental windows of sensitivity to social media, we first use cross-sectional data to examine whether adolescence might represent a period during which the association between well-being and social media is different in comparison to other life stages”. Age differences in patterns of association are confirmed by visual inspection of Figure 1. However, there are associations between social media use and life satisfaction across the lifespan, and as such it is not the case that adolescence is the only window of sensitivity. As such, the authors appear to be cherry picking the adolescent results (because this is the age period of interest) and ignoring the fact that there are associations in other age groups.

4.3 The new figure/analyses do not appear to take into account the nonlinear associations between social media use and life satisfaction, so it is unclear how this is a direct extension of the data presented in Figure 1.

4.4 I am still unclear whether the authors performed significance testing or only compared fit of models using AIC

4.6 The authors make two statements:

“as broader well-being measures are only available for ages 10-15 we cannot include them in our longitudinal analyses”.

“Thus more detailed analyses of such items are outside the scope of this manuscript”.

Why can't longitudinal analyses be performed for the age range 10-15? Why is such analysis outside the scope of the study? It seems very much in scope.

My final point is that language like “windows of sensitivity to social media” implies an effect of social media use on outcomes. This language should be tempered given the bidirectional effects observed. There is too much risk here of misinterpretation. The same can be said for “impacts of social media use” in the abstract. The abstract also seems to try and diminish the finding of an effect of life satisfaction on social media use by calling the effect “small” as compared to the opposite effect.

Reviewer #5 (Remarks to the Author):

I congratulate the authors on their thorough response to the reviewers' concerns.

I am happy that all my comments have been addressed satisfactorily.

Reviewer 1

The authors have adequately addressed all of my main criticisms in their revisions to the article and in their detailed and thorough response letter to the Reviewers.

Reviewer 2

Reviewer 2, Point 1: *I appreciate the changes that the authors have made. Clearly, my suggestions (reviewer 2) and the comments made by reviewer 1 in the previous round with respect to the use of causal language, were difficult to reconcile. While I still stand by my earlier suggestions that it would be helpful for psychology and related fields to be more open and upfront about one's interest in causal relations (even or especially in the absence of a randomized controlled trial), I understand the choices the authors have made and will not further push this matter.*

We thank the Reviewer for understanding the challenge of reconciling the mutually incompatible suggestion of making more concrete causal statements (Reviewer 2), while Reviewer 1 asked us to remove all causal statements. It was not possible to reconcile these two mutually exclusive requests and we therefore decided to adopt the more conservative of the two suggestions and remove causal statements from the paper. However, we value the Reviewer's contributions and are planning to apply them in future work, and added a section considering this issue to the manuscript: "First, to draw estimates of causal effects from these analyses one would need to adopt the following assumptions: a) there are no time-varying unobserved confounders that impact the relation between social media use and life satisfaction; b) that there is no measurement error in the variables; c) the time interval between studies (one year) is the right length to capture the effects of interest; and d) the bidirectional links estimated by our longitudinal model are linear in nature" (Line 302-307)

Reviewer 2, Point 2: *Line 35: "highlighted"; perhaps a better term in combination with "sensitivity" (which suggests a causal relation), is "suggested".*

We changed this accordingly: "Longitudinal analyses of 17,409 participants (10-21 years) suggested distinct developmental windows of sensitivity to social media in adolescence, when higher estimated social media use predicted a small decrease in life satisfaction one year later (and vice-versa: lower use predicted small increases in satisfaction)." (Line 34-28)

Reviewer 2, Point 3: *Lines 104-107: You indicate that the participants were measured up to 7 times; does this mean that the same individual is included multiple times in the data that were used to generate Fig 1, or was only one of these occasions used per person?*

We included the full data, and therefore multiple data points per participant if they were surveyed multiple times. We now add this to the caption: "Further, as most participants were measured

multiple times, more than one data point per participant will appear in this graph.” (Line 136-137)

Reviewer 2, Point 4: *Lines 142-143: “Goldilocks hypothesis”; I think a brief sentence, stating that this refers to the idea that too much or too little are both suboptimal, would be helpful (also for foreign readers that may not know this story or do not recognize the English name of it?).*

We have now included an explanation of the Goldilocks hypothesis: “This pattern in between-person associations, where those participants who use the least or the most social media also score lower in well-being, has been previously termed the ‘Goldilocks hypothesis’ (i.e., the concept that too much or too little digital technology use might be suboptimal).” (Line 144-147)

Reviewer 2, Point 5: *Lines 151-157, and elsewhere: You say you use AIC weights, but from the main text it is not really clear what this is and how it was used. I am not suggesting a major explanation should be given, but it would be helpful if some intuition is provided. Personally, I would have thought these are the same as Akaike weights, meaning that a set of models is compared and each model is given a probability of being the right (or best) model, and these probabilities add to 1. But in lines 183-187, these weights are used to present percentages, and these do not add up to 100, so I am not sure what the weights do and how they should be interpreted then (or how these percentages should be interpreted). Furthermore, I find the word “preferred” on p. 7 somewhat awkward; I wouldn’t say that the AIC (weights) prefer a particular model, but rather that we prefer a model based on the fit measures or weights, or that we select a model based on them; but this may be a matter of taste.*

Firstly, we have changed “AIC weights” to “Akaike weights”. We also now understand that our reporting was unclear. The percentages in 183-187 were from separate calculations of Akaike weights comparing two models: one with and one without sex differences. For example, “satisfaction with life 100%” means that the model with sex differences was 100% preferred over the models without sex differences. We have added this explanation to the text on Lines 188-191: “Akaike weight of model with sex difference compared to model without sex difference, i.e., the percentage shows how much more likely a model including a sex difference is likely to be the best model of the data compared to a model without a sex difference”

We have also changed the phrasing in lines 194: “Akaike weight of model with sex difference compared to model without sex difference”.

We have also changed the phrasing of the following sentences:

1. “This difference between males and females is statistically supported by an Akaike weights procedure³⁶ showing that models associating estimated social media use and life satisfaction while differentiating for self-reported sex (and controlling for household income, neighbourhood deprivation and year of data collection) are more likely to be the best models of the data between the ages of 12 and 15 (Akaike weights ratios ranging from 5:1 to over 6,000,000:1 in favour of the model differentiating males and females).” (Lines 155-160)
2. “In contrast, models differentiating for sex were not more likely to be the best models of the data at other ages (Figure 1, bottom; Extended Data Figure 3).” (Lines 160-162)
3. “Akaike weight of model with sex difference compared to model without sex difference, thus the percentage shows how much more likely a model including a sex difference is likely to be the best model of the data, compared to a model without a sex difference” (Lines 188-191)

4. “In the Millennium Cohort Study the models differentiating for sex were more likely to be the best model of the data for all measures” (Lines 192-193)
5. “A model constraining the path of life satisfaction predicting estimated social media use to be constant across age and sex, while allowing the path of estimated social media use predicting life satisfaction to vary, was more likely to be the best model for the data compared to other model constraints” (Lines 237-240)
6. “A Bayesian approach comparing a regression including age and the survey category as predictors of life satisfaction scores against a regression only including age as a predictor, found the latter model to be a better fit for the data with a Bayes Factor of 17.0 (+/- 2.1%).” (Lines 526-530)
7. “The Akaike weights procedure showed that the model constraining only life satisfaction predicting social media use was more likely to be the best model for the data (99.2%), while the model freeing both (0.7%), the model constraining social media use predicting life satisfaction (0.0%) and the model constraining both (0.1%) were not (Extended Data Figure 9).” (Lines 693-697)

Reviewer 2, Point 6: *Lines 299-300: “there are no time-varying covariates that impact the relation between social media use and life satisfaction” – I would be tempted to use the term “time-varying unobserved confounders”, to emphasize the causal nature of the effect of such a variable (although “impact” of course is also a causal term). Also, I think another assumption one needs to make here is that there is no measurement error in the variables.*

We have changed the text accordingly and have added the additional assumption of measurement error to the text

“First, to draw estimates of causal effects from these analyses one would need to adopt the following assumptions: a) there are no time-varying unobserved confounders that impact the relation between social media use and life satisfaction; b) that there is no measurement error in the variables” (Line 302-305)

Reviewer 2, Point 7: *Lines 322-324: “While the windows of sensitivity to social media are prominent in aggregate, they will most probably also meaningfully differ across individuals, as each person’s sensitivity is further influenced by a wide range of individual, peer and environmental dynamics.”; I would also say that a further line of research could be into what kind of social media use one is involved in (Facebook may have a different effect than twitter or TikTok), and how one uses it; for instance, there may be a difference in the impact based on whether one is more actively involved (e.g., posting and then seeing how many likes one gets), or more passively (e.g., mostly just looking/ reading and liking posts of others), and participating in WhatsApp groups may be different than using WhatsApp to communicate with individual friends.*

We thank the reviewer for this interesting suggestion and have added it to the text.

“Additionally, the type of social media use individuals are involved in will add further variance to this complicated dynamic.⁴⁹” (Line 330-332)

Reviewer 3

I thank the authors for their detailed response letter. They have satisfactorily responded to my previous concerns.

Reviewer 4

Reviewer 4, Point 1: 4.2. *In the manuscript, authors state: “To investigate the existence of developmental windows of sensitivity to social media, we first use cross-sectional data to examine whether adolescence might represent a period during which the association between well-being and social media is different in comparison to other life stages”. Age differences in patterns of association are confirmed by visual inspection of Figure 1. However, there are associations between social media use and life satisfaction across the lifespan, and as such it is not the case that adolescence is the only window of sensitivity. As such, the authors appear to be cherry picking the adolescent results (because this is the age period of interest) and ignoring the fact that there are associations in other age groups.*

The data we report reflect the original study designs, our developmental research questions and our interest and expertise in the adolescent period. The data were not selected to shape the conclusions arising from the findings we report. We have provided a detailed response to this point in Editor Point 1.

Reviewer 4, Point 2: 4.3 *The new figure/analyses do not appear to take into account the nonlinear associations between social media use and life satisfaction, so it is unclear how this is a direct extension of the data presented in Figure 1.*

We understand the concern about non-linear relations between social media use and life satisfaction across the age range. In Extended Data Figure 11 we fit the same model to the data four times: life satisfaction $\sim a*\text{social media} + b*\text{social media}^2$. However, the parameters a and b were either both allowed to vary by sex (“Free Both Terms”), only one was allowed to vary by sex (“Free Linear Term”/“Free Quadratic Term”) or none was allowed to vary by sex (“Constrained”). Looking at the figure, we can therefore examine whether there are quadratic trends in the data. What we find is that there is a lot of variation, but that the freed quadratic term is preferred, especially in the early adolescent age ranges.

In Extended Data Figure 3 we take the quadratic nature of the data into account to examine whether sex differences are present at different ages. In particular, the same model was fit to the data twice (life satisfaction $\sim a*\text{social media} + b*\text{social media}^2$), but one of the models allowed the parameters a and b to vary by sex, while the other model did not. The model fit indices (AIC) were compared using an Akaike weights procedure, and the ratio of the weights are plotted on the y-axis: the higher the bar the more a model with sex differences is preferred over a model without sex differences at that age group. The graph therefore takes into account non-linearity and shows that sex differences are most prominent in the early adolescent age range.

To further address Reviewer 4, Point 2, we have now updated Extended Data Figure 2 so that it also takes into account the non-linear associations between social media use and life satisfaction across the lifespan to understand whether there is a more negative relation between the two at different

ages. To produce the figure, we fit a quadratic regression model with covariates to the data, and then compare the predicted life satisfaction for those using 7 or more hours of social media and those using no social media (7+ hours – no social media). By adopting this approach, we now take into account the quadratic nature of the data, especially in older age groups. We have adapted the text and extended data in light of these analyses.

“Furthermore the finding was robust when adding control variables of income, neighbourhood deprivation (measured using the Index of Multiple deprivation) and year of data collection: adolescence was the only time the difference between the predicted life satisfaction of those using 7 or more hours of social media vs those using no social media, fell below -0.9 (modelled using a quadratic function; Extended Data Figure 2).” (Lines 112-117)

Extended Data Figure 2: Difference between the predicted life satisfaction of those participants using 7 or more hours of social media vs those using no social media

The graph plots the difference between the model predicted life satisfaction for those using 7+ hours of social media (score: 5) and those using no social media (score: 1) for different age groups (shown on the x axis). The model used predicted life satisfaction from social media use, social media use² and the covariates of household income, neighbourhood deprivation and year of data collection. It shows that the link between social media use and life satisfaction is most negative in early adolescence compared to older adolescence and adulthood.

Reviewer 4, Point 3: 4.4 I am still unclear whether the authors performed significance testing or only compared fit of models using AIC

Although for parts of the models we also report the significance of certain parameters, for the purposes of model comparison we have consistently used Akaike weights. These offer a benefit over LRT comparisons in terms of model symmetry: A likelihood ratio test can only fail to favour the more complex model, leading to a non-significant associated p-value. In contrast, Akaike weights (as

do other procedures) allow us to compare the evidence in favour of either model symmetrically: In other words, model comparison may show that the simpler model is the more likely model given the data, and the strength of such an inference can be (unlike non-significant p-values) be interpreted accordingly. Given the large dataset we wanted to ensure that our model selection procedure offered the opportunity to favour a simpler model, rather than solely be focused on statistical significance.

Reviewer 4, Point 4: *4.6 The authors make two statement: “as broader well-being measures are only available for ages 10-15 we cannot include them in our longitudinal analyses”. “Thus more detailed analyses of such items are outside the scope of this manuscript”. Why can’t longitudinal analyses be performed for the age range 10-15? Why is such analysis outside the scope of the study? It seems very much in scope.*

As explained in our response to Editor Point 1 (and Reviewer 4 Point 1,) we believe the most important and plausible developmental changes in the link between social media use and life satisfaction would occur over shorter time scales. The depressive symptoms, self-esteem, externalising, and internalising measures were only completed by 10-15-year-olds every two years (in Understanding Society; MCS data was cross-sectional only). We therefore did not apply longitudinal models as these would have tested whether social media use in one year impacts the mental health measures (e.g., self-esteem) *two years later* or vice versa. This theoretical and methodological challenge has been discussed at length in the developmental literature (Adolph et al., 2008, Dormann and Griffin, 2015).

However, the extended life satisfaction questions shown in Figure 2 (e.g., satisfaction with friends) were completed annually by 10-15-year-olds, and we have added longitudinal analyses including these questions into revised supplementary materials, thereby integrating this into our updated manuscript (the Figure is shown in our response to Editor Point 1 on page 3).

Reviewer 4, Point 5: *My final point is that language like “windows of sensitivity to social media” implies an effect of social media use on outcomes. This language should be tempered given the bidirectional effects observed. There is too much risk here of misinterpretation. The same can be said for “impacts of social media use” in the abstract. The abstract also seems to try and diminish the finding of an effect of life satisfaction on social media use by calling the effect “small” as compared to the opposite effect.*

We have done various edits to the abstract to address these comments as we in no way favour an effect in either direction. We have revisited the manuscript to ensure consistency of reporting throughout, as we agree with the reviewer that such asymmetry would not be defensible.

4. We use the term windows of sensitivity to describe the age-dependent changes in the predictive link between social media use and life satisfaction one year later. No age-dependent differences were found in the opposite path (life satisfaction predicting social media use one year later). While this path was significant, and we report it as such, it did not change by age. We therefore do not think we are misleading readers in our reference to “windows of sensitivity to social media”.
5. We used “impacts of social media use” in the last sentence of the abstract to reference future research, not the current study. We now understand this was unclear and have changed it accordingly: “The findings highlight that adolescent development needs to be considered in social media use research.” (Line 41-43)

6. We have deleted the word “small” to ensure it does not look like we are misrepresenting the results found for life satisfaction predicting social media use, and we further clarified that this path was not age-dependent: “Decreases in life satisfaction also predicted subsequent increases in social media use, however these were not impacted by age”. (Line 40-41)
7. In response to Reviewer 2, Point 2, we also changed the word “highlighted” to “suggested”, further tempering the abstract: “Longitudinal analyses of 17,409 participants (10-21 years) suggested distinct developmental windows of sensitivity to social media in adolescence, when higher estimated social media use predicted a small decrease in life satisfaction one year later (and vice-versa: lower use predicted small increases in satisfaction).” (Line 34-28)

Reviewer 5

I congratulate the authors on their thorough response to the reviewers' concerns.

I am happy that all my comments have been addressed satisfactorily.

Summary

We thank the Editor and reviewers for taking the time to consider our manuscript. Based on these considerations, I'm writing to ask if you would be willing to reconsider your decision and allow us to revise our paper, which we are confident could be achieved on an efficient timeline. Please do not hesitate to contact me for any further clarification.

Yours Sincerely,

Amy Orben

College Research Fellow

Emmanuel College & MRC Cognition and Brain Sciences Unit

University of Cambridge

REVIEWER COMMENTS

Reviewer #2 (Remarks to the Author):

I appreciate the changes the authors have made in response to my review. I believe this is very interesting paper, and it reads very well.

I have a few very minor points (mostly just edits):

- In the Figure captions of Figures 1 and 2, there are statements about confidence intervals; however, these are not included in these plots (anymore)
- In the Figure caption of Figure 3, there is a statement about grey boxes, thin lines, and dark and light shaded ribbons; but these were not visible in the plot
- I am not a native speaker, but I have the impression that one of the "likely"'s in the sentences to describe the Akaike weights should/could be dropped
- Regarding the causal interpretation of parameters in the RI-CLPM, I realize that what should be added as a nuance is that one needs to assume that the model adequately accounts for unobserved time-invariant confounding through the inclusion of the random intercept (i.e., when the effect of unobserved time-invariant confounders changes over time, this may not be adequately captured through this latent factor with all loadings set to 1).

Reviewer #4 (Remarks to the Author):

I believe that the authors misunderstood my comment regarding "cherry picking". My point was that there appear to be associations between social media use and life satisfaction during age periods other than early adolescence. Even if adolescence is the period the authors are interested in, it seems strange to me to not discuss/interpret findings at other ages. For example, a similar association was found at age 28-29 (albeit in males only), but this was not discussed. While the authors claim that associations were "different" during adolescence, it is unclear what this means.

On page 5, the authors state "adolescence was the only time the difference between the predicted life satisfaction of those using 7 or more hours of social media vs those using no social media, fell below - 0.9". Please explain further - readers are not going to know the significance of "-0.9".

In the legend of Figures 1/2/3, confidence intervals are mentioned, but these are not visible.

In Extended Data Figure 7, it seems that life satisfaction (some domains) prospectively predict social media use (in addition to vice versa). This result does not appear to be discussed in the main paper, but I believe that it should.

15 November 2021

Reviewer 2

Reviewer 2, Point 1: *I appreciate the changes the authors have made in response to my review. I believe this is very interesting paper, and it reads very well. I have a few very minor points (mostly just edits). In the Figure captions of Figures 1 and 2, there are statements about confidence intervals; however, these are not included in these plots (anymore).*

We have checked the submission proofs and the reviewer is correct that a lot of plot detail was removed from the plots at point of uploading to the submission website (the original word document has the full figures included). We will now ensure that the figure remains uncorrupted during our resubmission.

Reviewer 2, Point 2: *In the Figure caption of Figure 3, there is a statement about grey boxes, thin lines, and dark and light shaded ribbons; but these were not visible in the plot.*

See response to Reviewer 2, Point 1.

Reviewer 2, Point 3: *I am not a native speaker, but I have the impression that one of the "likely"s in the sentences to describe the Akaike weights should/could be dropped.*

We agree and have changed this accordingly.

“In Understanding Society, sex differences were found predominately for satisfaction with life and appearance (Akaike weight of model with sex difference compared to model without sex difference, i.e., the percentage shows how much more likely a model including a sex difference is the best model of the data compared to a model without a sex difference: satisfaction with life 71.9%, appearance 69.1%, family 53.8%, friends 59.1%, school 51%, schoolwork 35%).” (Lines 226-231)

Reviewer 2, Point 4: *Regarding the causal interpretation of parameters in the RI-CLPM, I realize that what should be added as a nuance is that one needs to assume that the model adequately accounts for unobserved time-invariant confounding through the inclusion of the random intercept (i.e., when the effect of unobserved time-invariant confounders changes over time, this may not be adequately captured through this latent factor with all loadings set to 1).*

We have added this assumption to the relevant section.

“First, to interpret the parameters from our analyses as estimates of causal effects one would need to adopt the following assumptions: a) there are no time-varying unobserved confounders that impact the relation between social media use and life satisfaction; b) the model adequately accounts for unobserved time-invariant confounding through the inclusion of a random intercept; c) there is no measurement error in the variables; d) the time interval between studies (one year) is the right length to capture the effects of interest; and e) the bidirectional links estimated by our longitudinal model

are linear in nature.” (Lines 371-378)

Reviewer 4

Reviewer 4, Point 1: *I believe that the authors misunderstood my comment regarding "cherry picking". My point was that there appear to be associations between social media use and life satisfaction during age periods other than early adolescence. Even if adolescence is the period the authors are interested in, it seems strange to me to not discuss/interpret findings at other ages. For example, a similar association was found at age 28-29 (albeit in males only), but this was not discussed. While the authors claim that associations were "different" during adolescence, it is unclear what this means.*

We have restructured our descriptions on page 5 to describe and interpret the findings across the different age ranges more accurately and explain why we focus on adolescence in the main parts of the paper.

“To address the first research question, we analysed the UK Understanding Society household panel survey that spans 72,287 10-80 year old participants surveyed up to seven times each between 2011 and 2018³⁴, correlating a single-item life satisfaction measure and participant estimates of how much time they spend using social media on a typical day (raw data plot: Figure 1; extended plot: Extended Data Figure 1). We also tested the robustness of these relations using – both linear and quadratic – terms of social media use to predict life satisfaction while adding control variables of income, neighbourhood deprivation (measured using the Index of Multiple deprivation) and year of data collection (Extended Data Figure 2).

While it is important to note that responses to, and conceptualizations of, life satisfaction might be qualitatively different across the lifespan, our results showed that the relationship between estimated social media use and life satisfaction varied substantially by age. Although the relationship fluctuates to a certain extent across the lifespan, for example it is more negative in males aged 26-29 years, the most substantial negative relations were found in adolescence (Extended Data Figure 2). This finding, combined with our *a priori* developmental interest in adolescence, the fact that social media use is heightened in this age group and the nature of the data (annual social media measures being available only for those aged 10-21), motivated us to focus on adolescence as the age group of interest throughout this study.” (Lines 120-140)

We also added some additional description of how there are other ages where the relationship between social media use and life satisfaction became more negative in the caption of Extended Data Figure 2:

“It shows that the link between social media use and life satisfaction is most negative in early adolescence compared with older adolescence and adulthood, but there are other ages when the relationship also becomes slightly more negative (e.g., males aged 26-29).”

We also changed the phrasing on pages 7 and 8, so that we do not state that adolescence is the only time the link between social media use and life satisfaction was more negative.

“These analyses demonstrated that the between-person association linking estimated social media use to life satisfaction was more negative in adolescents compared with most other age groups. Further, they showed that adolescence is unique due to the prominent sex differences on the cross-sectional links between estimate social media use and life satisfaction that are not evident at most other ages.” (Lines 198-202)

Lastly, in the discussion, we interpret the lifespan findings and call for increased research in this area.

“Furthermore, the cross-sectional relation between social media use and life satisfaction showed differences across the whole life span, e.g., in early adulthood and old age. Future work may use a similar approach to investigate interactions in older age groups with a suitably rich sample.” (Lines 394-397)

Reviewer 4, Point 2: *On page 5, the authors state "adolescence was the only time the difference between the predicted life satisfaction of those using 7 or more hours of social media vs those using no social media, fell below -0.9". Please explain further - readers are not going to know the significance of "-0.9".*

We understand this concern and have changed our explanation of Extended Data Figure 2 in the main text.

“Although the relationship fluctuates to a certain extent across the lifespan, for example it is more negative in males aged 26-29 years, the most substantial negative relations were found in adolescence (Extended Data Figure 2).” (Lines 132-136)

Reviewer 4, Point 3: *In the legend of Figures 1/2/3, confidence intervals are mentioned, but these are not visible.*

We have checked the submission proofs and the reviewer is correct that a lot of plot detail was removed from the plots at point of uploading to the submission website (the original word document has the full figures included). We will now ensure that the figure remains uncorrupted during our resubmission.

Reviewer 4, Point 4: *In Extended Data Figure 7, it seems that life satisfaction (some domains) prospectively predict social media use (in addition to vice versa). This result does not appear to be discussed in the main paper, but I believe that it should.*

We understand Reviewer 4’s repeated concern that the path of life satisfaction prediction social media use is not emphasised enough in the manuscript. To address this, we have now moved Extended Data Figure 8 into the main manuscript (Figure 3). It shows both paths (social media predicting life satisfaction and vice versa), ensuring that this bidirectionality is emphasised visually. This is further complemented by previous revisions to our manuscript: e.g, changes to the text and the inclusion of Table 1.

Figure 3: Results from Random Intercept Cross-Lagged Panel Model (RI-CLPM) of estimated social media use and life satisfaction for 17,409 participants of the Understanding Society dataset aged 10-21 (52,556 measurement occasions).

Results from both cross-lagged paths of a RI-CLPM where those paths were free to vary across age/sex. Results are unstandardised and split by path (left: deviations from expected life satisfaction at that age predicting deviations from expected social media use one year later; right: deviations from expected social media use at that age predicting deviations from expected life satisfaction one year later) and sex (female = top/red, male = bottom/blue). The ribbon represents the 95% Confidence Interval. (Lines 275-287)

We also have included a sentence explaining that our focus on the path linking social media use to life satisfaction (due to its age and sex variations), does not mean the opposite path is unimportant.

“We will focus on these differences in the next section, but wish to emphasize that this focus does not mean that the path linking life satisfaction in one year to social media use one year later is unimportant.” (Lines 313-315)

REVIEWERS' COMMENTS

Reviewer #4 (Remarks to the Author):

The authors have addressed my remaining concerns. Thank you.